# CAN REINFORCEMENT LEARNING SOLVE ASYMMETRIC COMBINATORIAL-CONTINUOUS ZERO-SUM GAMES?

**Yuheng Li, Panpan Wang & Haipeng Chen**
Department of Data Science
College of William & Mary
{yli95,pwang12,hchen23}@wm.edu

## ABSTRACT

There have been extensive studies on learning in zero-sum games, focusing on the analysis of the existence and algorithmic convergence of Nash equilibrium (NE). Existing studies mainly focus on symmetric games where the strategy spaces of the players are of the same type and size. For the few studies that do consider asymmetric games, they are mostly restricted to matrix games. In this paper, we define and study a new practical class of asymmetric games called two-player Asymmetric Combinatorial-Continuous zEro-Sum (ACCES) games, featuring a combinatorial action space for one player and an infinite compact space for the other. Such ACCES games have broad implications in the real world, particularly in combinatorial optimization problems (COPs) where one player optimizes a solution in a combinatorial space, and the opponent plays against it in an infinite (continuous) compact space (e.g., a nature player deciding epistemic parameters of the environmental model). Our first key contribution is to prove the existence of NE for two-player ACCES games, using the idea of essentially finite game approximation. Building on the theoretical insights and double oracle (DO)-based solutions to complex zero-sum games, our second contribution is to design the novel algorithm, Combinatorial Continuous DO (CCDO), to solve ACCES games, and prove the convergence of the proposed algorithm. Considering the NP-hardness of most COPs and recent advancements in reinforcement learning (RL)-based solutions to COPs, our third contribution is to propose a practical algorithm to solve NE in the real world, CCDORL (based on CCDO) and provide the novel convergence analysis in the ACCES game. Experimental results across diverse instances of COPs demonstrate the empirical effectiveness of our algorithms. The code of this work is available at `https://github.com/wmd3i/CCDO-RL`.

## 1 INTRODUCTION

Zero-sum games depict a game theoretic paradigm among adversarial players, where the increase in one player's rewards inevitably leads to a decrease in the other's (Lipton and Young, 1994). It is prevalent in various real-world domains such as board games (Ghory, 2004), poker (Zinkevich et al., 2007), and price games (Kakkar et al., 2022). Since solving NE plays a vital role in the game theory, from fictitious play (Brown, 1951), and double oracle (DO) (McMahan et al., 2003) to Policy-Space Response Oracles (Lanctot et al., 2017), numerous algorithms have endeavored to find NE while providing a theoretical analysis of algorithmic convergence and approximation (Jafari et al., 2001; Waugh and Bagnell, 2015; Balandat et al., 2016; Dinh et al., 2022; Tang et al., 2023).

One way to classify zero-sum games is based on the symmetry of the players' strategy spaces (Amir et al., 2008; Cox et al., 2013; Stella and Bauso, 2018). A *symmetric* game describes a scenario where players can be interchanged (Cheng et al., 2004), meaning that all the players have the same strategy set and payoff matrices. Otherwise, the game is *asymmetric*. Symmetric games have been well-studied in terms of both theories (Tuyls et al., 2018b; Hefti, 2017) and applications (Bichler et al., 2021; Altman et al., 2011), partly due to their simple and structural properties. However, in many scenarios such as Leduc Poker (Tuyls et al., 2018a), network security games (Wilder,

2018), and cash-in-transit VRP (Ghannadpour and Zandiyeh, 2020), the strategy spaces of the players are asymmetric.

Despite the extensive literature on asymmetric games, most current studies remain confined to relatively traditional backgrounds such as the Battle of the Sexes game (Tuyls et al., 2018b) and Leduc Poker (Tuyls et al., 2018a). As a kind of strategy space that is ubiquitous in real-world applications, much less exploration has been made toward asymmetric games with combinatorial strategy spaces, except for some sporadic studies like min-max traveling distance of multi-VRP (Narasimha et al., 2013), security scheduling with attacker (Jain et al., 2011), max-min influence maximization (Chen et al., 2016), etc. Typically, these studies assume finite action spaces for all players in asymmetric game settings. These studies neglect another broad class of asymmetric games where the other player's strategy space is not only asymmetric but also infinite compact (e.g., real-valued vector intervals). Such infinite compact strategy spaces in asymmetric games have broad implications in the real world, which can be interpreted as the physical or environmental parameters of COPs, such as the attractive degrees to targets in the security game (Xu et al., 2021), uncertain network edge weights in influence maximization problems (Kalimeris et al., 2019) and unknown outer condition effect on the charging demand in facility location problems (An, 2020; Tirkolaee et al., 2020), and customer demand in routing problems (Florio et al., 2023).

Formally, we define this class of games as a two-player Asymmetric Combinatorial-Continuous zEro-Sum (ACCES) game with dynamics of simultaneous move and static form. Player 1's strategy space is combinatorial, while Player 2's is infinite and compact with a continuous utility function. As an illustrative example (more examples in Section 6), we consider a patrolling game between a defender (Player 1) and an attacker (Player 2). To prevent attacks from the attacker, the defender chooses a feasible route to patrol a subset $\Pi$ of all targets $\{1, 2, \ldots, N\}$ meanwhile satisfying the total distance constraint $L_{all}$ because of limited patrol time. For the attacker, the strategy is the attack probability vector $\{p_1, p_2, \ldots, p_N\}$ for the target set. Besides, each target $i \in \{1, 2, \ldots, N\}$ has its own value $v_i$. The utility function for the defender is the expectation of successfully protected target values, i.e. $U_d = \sum_{i=1}^{N} p_i v_i \mathbb{I}_\Pi$. The attacker's utility function is then: $U_a = -U_d$.

For this new class of games, our key research question is:

*"Whether and how can we solve asymmetric combinatorial-continuous zero-sum games?"*

This question can be decomposed into the following sub-questions:

**1) Does NE exist?** Before finding solutions to ACCES games, the first question is whether such games are guaranteed to have NE. Due to the asymmetry of the game, especially the different, less-structured properties of the strategy sets (combinatorial-continuous), this question is much less straightforward than established results in matrix games (Nash et al., 1950), market games (Peck et al., 1992), and continuous games (Glicksberg, 1952; Fan, 1952; Reny, 2005), for which the existence of NE has already been proven.

**2) Is there any algorithm that can converge to NE?** If the existence of NE holds, the next question is then to find an algorithm that can converge to the NE. Due to the infinite strategy set of one player, common equilibrium-seeking algorithms (McMahan et al., 2003; Dinh et al., 2022) in matrix games lose their convergence guarantees because they rely on the finiteness of strategy sets to terminate iterations. On the other hand, classic algorithms in continuous games (Balandat et al., 2016; Adam et al., 2021) do not work for ACCES games because the continuity of the unity function no longer holds with the discrete (combinatorial) strategy set of the other player.

**3) Is there a practical algorithm that we can actually implement in the real world?** While it is critical to understand the theoretical questions above, an equally important practical question is – how can we design efficient and practical algorithms to actually solve the ACCES games? This is extremely challenging as even a sub-problem of finding the best response for the combinatorial strategy space of one player is known to be NP-hard, let alone the entire ACCES game.

We give a YES answer to each of the three sub-questions. Our main contributions are as follows.

1) We are the first to summarize and define the class of ACCES games, elucidating its rationale and practical significance via examples from min-max games and security games.

2) We prove the existence of mixed NE in ACCES games through the finite-game approximation, which relies on two important properties that have yet to be established, the weakly sequential com-

pactness and continuity of expected utility function. To address this gap, we prove these properties in Section 4, which provides further insight into developing solution algorithms for ACCES games.

3) We propose two solution algorithms, CCDO and CCDO-RL, for solving the ACCES games. CCDO extends the idea of double oracle (DO)-based solutions from zero-sum finite games (McMahan et al., 2003) to ACCES games, while with different convergence guarantee results. Due to the NP-hardness in most COPs, it is infeasible to find the exact best response for the combinatorial player in a limited time. Therefore it's critical to consider the solution algorithm and convergence analysis with approximate best responses (ABRs). We bridge this gap by proposing CCDO-RL which adopts RL as an efficient sub-routine to compute the ABRs, inspired by recent advancements in applying reinforcement learning to learn fast, effective, and generalizable solutions for COPs (Khalil et al., 2017; Nazari et al., 2018; Deudon et al., 2018; Bengio et al., 2020; Chen et al., 2021; Berto et al., 2023). Furthermore, novel convergence analysis of CCDO-RL is studied, along with different ABRs' influence on convergence have been discussed in Section 5.

4) We validate our algorithms on three distinct instances of ACCES games – adversarial covering salesman problem (ACSP), adversarial capacitated vehicle routing problem (ACVRP), and patrolling game (PG). Empirical results are well aligned with our theoretical insights: our proposed CCDO-RL algorithm can learn to converge to NE in all game instances. For the player with the combinatorial strategy space, our algorithm is better than baselines regardless of the type of adversary or problem size, especially in terms of generalizability.

## 2 RELATED WORK

**Symmetric and asymmetric games.** Symmetric games are initially proposed by (Von Neumann and Morgenstern, 1947) and studied under the context of non-cooperative (Nash et al., 1950), economic (Hammerstein and Selten, 1994), and two-person (Washburn et al., 2014) games. Amir et al. (2008) focus on pure strategy equilibrium with supermodular payoff functions. Fey (2012) studies symmetric games only with asymmetric equilibria. A few studies extend the theories on symmetric games to asymmetric settings (Cox et al., 2013; Tuyls et al., 2018a), or transform asymmetric games to symmetric ones (Tuyls et al., 2018a). These studies usually concentrate on a specific type of classic game such as metric games or poker. Narasimha et al. (2013) and Carlsson et al. (2009) are among the first to consider asymmetric games involving a combinatorial player in a variant of traveling salesman problem with multiple vehicles. Jain et al. (2011) studies a security game where the defender decides the location of sources and the attacker chooses a path to find the source. Xu et al. (2014) extends the idea of Jain et al. (2011) and consider discrete time domains and moving targets. In the covering problem, Rahmattalabi et al. (2019) considers the failure of some nodes and models it as one zero-sum game. All of these studies are limited to finite games.

**Equilibrium learning in zero-sum games.** DO (McMahan et al., 2003) is a powerful tool for solving complex strategic normal-form games by iteratively expanding the players' strategy sets and efficiently finding equilibria. The idea has been extended toward better NE computation, different forms of games, convergence rate, etc. McAleer et al. (2020) and Zhou et al. (2023) focus on accelerating the computation of the approximate equilibrium. Different diversity metrics are proposed by (Balduzzi et al., 2019; Perez-Nieves et al., 2021; Liu et al., 2021; Yao et al., 2024) to find more effective and various strategies. In extensive-form games, McAleer et al. (2021) works to achieve linear convergence to approximate equilibrium and Tang et al. (2023) studies sample complexity. Except for DO and its variants, NE learning in zero-sum settings remains appealing in periodic games (Fiez et al., 2021), polymatrix games (Cai et al., 2016), and Markov games (Zhu and Zhao, 2020), etc. As far as we know, they are all limited to matrix games in theories related to the existence and convergence of NE although McAleer et al. (2021) conduct experiments on continuous-action games by Deep RL. Balandat et al. (2016); Adam et al. (2021) study the NE convergence of continuous games but two players are symmetric.

**RL for COPs.** RL has emerged as an effective and generalizable method to solve COPs, where the underlying idea is to decompose the original combinatorial action selection in COPs into a sequence of greedily selected individual actions, using a deep RL policy or value function that is usually represented via various function approximation methods such as graph neural networks (Khalil et al., 2017; Joshi et al., 2019; Manchanda et al., 2020), recurrent (Bello* et al., 2017), and attention net-

works (Kool et al., 2018). Search algorithms, such as active search (Hottung et al., 2022), Monte Carlo tree search (Fu et al., 2021), and multiple rollouts (Kwon et al., 2020), are further integrated into these frameworks to enhance the solution qualities of RL algorithms during inference time. Integrating representation learning and search algorithms, RL has shown promising abilities to learn efficient and generalizable solutions to complex COPs. This motivates us to adopt RL as the backbone method to compute the COPs in a subgame of the ACCESS games.

## 3 PRELIMINARIES

### 3.1 TWO-PLAYER ASYMMETRIC COMBINATORIAL-CONTINUOUS ZERO-SUM (ACCES) GAMES

Most two-player zero-sum strategic games are described as a payoff matrix $\Pi_{n \times m}$ where the rows and columns represent pure strategies for the two players. This does not hold for ACCES games because the strategy space for the continuous player is infinite. Hence, we provide the first formal formulation of ACCES games.

Formally, we represent a two-player ACCES game a tuple $\{X, Y, u\}$, where $X$ is the combinatorial but finite space, and $Y$ is a compact and infinite metric space, as the pure strategy space for players 1 and 2 respectively. $u$ is the utility function mapping the joint strategy space $X \times Y$ to a scalar $\mathbb{R}$, with the continuity on $Y$ when fixing $x \in X$. The utility function of Player 1 is $u$, and for Player 2 is $-u$. For the security patrolling game exemplified in the introduction, $X$ should be all routes that satisfy the distance constraint, $Y$ is the real vector interval $[0, 1]$ for the attack probability $p_i$ on each target $i = 1, ..., N$, and $u$ is the expectation of successfully attacked target negative values.

The mixed strategy in the combinatorial-continuous game is defined separately because two players own entirely different forms of strategy spaces. For Player 1, the set of *mixed strategies* can be written as $\triangle_X \triangleq \{p = [p(x_1), ..., p(x_{|X|})] | \sum_{i=1}^{|X|} p(x_i) = 1, p(x_i) \geqslant 0\}$, where $p(x_i)$ is regarded as the chosen probability of the pure strategy $x_i \in X$. For Player 2, a mixed strategy is a Borel probability measure $q$ on $Y$ which can be seen as a probability distribution function $q : \mathcal{F} \rightarrow [0, 1]$, where $\mathcal{F}$ is $\sigma$- algebra of $Y$. The set of mixed strategies of Player 2 is denoted by $\triangle_Y$. Every mixed strategy in $\triangle_X$ corresponds to a distribution on all feasible routes in the security patrolling game, and that in $\triangle_Y$ passes as a cumulative distribution function defined on $[0, 1]^N$.

Due to the infiniteness of strategy space $\mathcal{Y}$, the support of $q$ may be infinite. Given a mixed strategy $(p, q) \in \triangle_X \times \triangle_Y \triangleq \triangle$, the *expected utility function* of Player 1 can be defined as

$$U(p, q) = \sum_{x \in X} \int_{y \in Y} p(x) u(x, y) dq. \tag{1}$$

Correspondingly, the expected utility function of Player 2 is $-U(p, q)$.

### 3.2 NASH EQUILIBRIUM IN TWO-PLAYER ACCES GAMES

In two-player ACCES games, a mixed strategy pair $(p^*, q^*)$ is *Nash equilibrium (NE)* if and only if
$$U(p, q^*) \leqslant U(p^*, q^*) \leqslant U(p^*, q), \forall p \in \triangle_X, q \in \triangle_Y. \tag{2}$$

Additionally, we denote $\epsilon$- *NE* as a mixed strategy pair $(p^*, q^*)$ which satisfies
$$U(p, q^*) - \epsilon \leqslant U(p^*, q^*) \leqslant U(p^*, q) + \epsilon, \forall p \in \triangle_X, q \in \triangle_Y. \tag{3}$$

*Best response* $\mathbb{BR}_i(\pi_{-i})$ defines the best pure strategy for Player $i$ for a fixed mixed strategy $\pi_{-i}$ of the other player $-i$. In ACCES games, the set of best responses for the two players are:
$$\mathbb{BR}_1(q) = \{x \in X | U(x, q) = \max_{x' \in X} U(x', q)\}, \mathbb{BR}_2(p) = \{y \in Y | U(p, y) = \min_{y' \in Y} U(p, y')\}. \tag{4}$$

In many situations, finding the best response is inherently difficult, especially in most combinatorial optimization problems which are $NP$-hard. Approximate or heuristic algorithms are often used to sacrifice solution accuracy for faster computation. We use $\epsilon$- *best response* $\mathbb{BR}_i^\epsilon(\pi_{-i})$ to define the solution that is no worse than the ground truth best response by $\epsilon$:
$$\mathbb{BR}_1^\epsilon(q) = \{x \in X | U(x, q) \geqslant \max_{x' \in X} U(x', q) - \epsilon\}, \mathbb{BR}_2^\epsilon(p) = \{y \in Y | U(p, y) \leqslant \min_{y' \in Y} U(p, y') + \epsilon\}. \tag{5}$$

## 4 THE EXISTENCE OF NASH EQUILIBRIUM

We first study the existence of Nash equilibrium in ACCES games, which is a critical step before designing any actual solutions. For a two-player ACCES game $\mathcal{G} = \{X, Y, u\}$, the strategy set $X$ of Player 1 is finite consisting of certain permutations/combinations of nodes, although the number of the strategy set is possibly exponentially large. In contrast, the strategy set $Y$ of Player 2 is an infinite and compact set. Although the utility function of Player 2 are continuous on $Y$ when fixing $x \in X$, but its strategy's infiniteness disqualifies the finite condition of matrix games and makes the convergence to NE less straightforward. Meanwhile, the discreteness of $X$ destroys the continuity of the utility function on $X \times Y$. To see whether the existence of mixed strategy NE still holds in ACCES games, we must better understand the game structure. Our thought flow is as follows.

On a high level, we first prove Proposition 1 of weakly sequential compactness in the mixed strategy product space of ACCES games. Then, the continuity of the expected utility function on the product space, which contributes to the existence proof and the following convergence of algorithms in Section 5, is proven in Proposition 2. Note that these are two key technical novelties that not only are critical intermediate steps for the proof of the existence of NE, but also build the foundation of the analysis of convergence to NE of our proposed algorithms in Section 5.1.

**Proposition 1** [Weakly Sequential Compactness.] *Set the ACCES game is $\mathcal{G} = (X, Y, u)$, where $X$ is finite, $Y$ is a nonempty compact metric space, and the utility function $u$ is continuous on $Y$ fixing $x \in X$. Then the joint mixed strategy space $\triangle \triangleq \triangle_X \times \triangle_Y$ is weakly sequentially compact.*

*Proof sketch.* To prove the product space $X \times Y$ is weakly sequential compact, we just need to prove two parts, weakly sequential compactness and separability of $X, Y$ based on Lemma 1. See the full proof in Appendix A.1. □

**Proposition 2** [Continuity of Expected Utility Function.] *The expected utility function $U(p, q) \triangleq \sum_{x \in X} \int_{y \in Y} p(x)u(x, y)dq$ is continuous on the joint mixed strategy space $\triangle$, $\forall p \in \triangle_X, q \in \triangle_Y$.*

*Proof sketch.* We prove the continuity of the expected utility function by definition. First, define the metric distance on mixed strategy sets $\triangle_X, \triangle_Y$ and their product space $\triangle_X \times \triangle_Y$. Following this, the distance between two mixed strategy pairs $(p, q)$ and $(p', q')$ can be scaled to the distance sum between $p, p'$ and $q, q'$ because of the compactness of $Y$, the continuity of utility function on $Y$, and Proposition 1. The full proof is provided in Appendix A.1. □

Via Proposition 2 and the continuity of $U$ on $Y$, the following two statements hold:

- When $p_n \Rightarrow p$ in $\triangle_X$, $q_n \Rightarrow q$ in $\triangle_Y$, $U(p_n, q_n) \to U(p, q)$.
- When $p_n \Rightarrow p$ in $\triangle_X$, $y_n \to y$ in $Y$, $U(p_n, y_n) \to U(p, y)$.

Secondly, for the proof of equilibrium existence, we build on the idea in (Myerson, 1991) which approximates the strategy spaces by finite grids. To describe the approximation and the feasibility of approximation by finite games, we first introduce definitions of $\alpha$-approximate games and essentially finite games. Based on these definitions, we establish Propositions 3, and 4, where the proofs are provided in Appendix A.1.

**Definition 1** [$\alpha$-Approximate Game.] Assume there exist two strategic games $\mathcal{G} = \langle X, Y, u \rangle, \mathcal{G}' = \langle X, Y, u' \rangle$ in which $u$ and $u'$ are bounded and measurable utility functions. If every joint strategy $(x, y) \in X \times Y$, $|u(x, y) - u'(x, y)| \leqslant \alpha$, then $\mathcal{G}'$ is an $\alpha$-approximation of $\mathcal{G}$.

**Definition 2** [Essentially Finite Game.] The game $\mathcal{G} = \langle X, Y, u \rangle$ is essentially finite, if and only if there exists some finite strategic game $\hat{\mathcal{G}} = \langle \hat{X}, \hat{Y}, \hat{u} \rangle$ and measurable functions $f_1 : X \to \hat{X}$, $f_2 : Y \to \hat{Y}$ s.t. $u(x, y) = \hat{u}(f_1(x), f_2(y)), \forall (x, y) \in X \times Y$.

**Proposition 3** [Approximate NE of ACCES.] *$\mathcal{G}' = \langle X, Y, \tilde{u} \rangle$ is $\alpha$-approximation of $\mathcal{G} = \langle X, Y, u \rangle$, where $\mathcal{G}$ is an ACCES game. $(p^*, q^*)$ is an $\epsilon$-equilibrium of $\mathcal{G}'$, then $(p^*, q^*)$ is an $(\epsilon + 2\alpha)$-equilibrium of $\mathcal{G}$.*

**Proposition 4** [Essentially Finite of ACCES.] *For an ACCES $\mathcal{G}$, $\forall \alpha > 0$, there exists an essentially finite strategic game $\hat{\mathcal{G}} = \langle X, \hat{Y}, \hat{u} \rangle$, s.t. $\hat{\mathcal{G}}$ is $\alpha$-approximation of $\mathcal{G}$.*

**Proposition 5** [Convergence of Approximate ACCES NE.] $\mathcal{G}$ *is an ACCES game, for each* $n$, $(p_n, q_n)$ *is* $\epsilon_n$-NE *of* $\mathcal{G}$, $(p_n, q_n) \Rightarrow (p^*, q^*)$, $\epsilon_n \to \epsilon$, *then* $(p^*, q^*)$ *is an* $\epsilon$-*equilibrium of* $\mathcal{G}$.

Based on Chapter 3 in Myerson (1991) and Proposition 2, Proposition 5 holds naturally. On account of Proposition 3, 4, and 5, the existence of equilibrium can be obtained. We have provided a further discussion of the existence of NE for $N$-player ACCES games in Appendix A.2.

**Theorem 1 [Existence of NE]** $\mathcal{G} = \langle X, Y, u \rangle$, *where* $X$ *is finite space,* $Y$ *is nonempty compact metric space,* $u : X \times Y \to \mathbb{R}$ *is a continuous utility function on* $Y$ *when fixing* $x \in X$. *Game* $\mathcal{G}$ *has a mixed strategy Nash equilibrium.*

*Proof sketch.* For any sequence $\{\alpha_k\} \to 0$, there exists an essentially finite game sequence $\{\mathcal{G}_k\}$ (Prop 4) such that its NE $\{(p_k^*, q_k^*)\}$ is $2\alpha_k$- NE of the initial game $\mathcal{G}$ (Prop3). We can prove that the sequence $\{(p_k^*, q_k^*)\}$ converges and its convergent point $(p^*, q^*)$ is the NE of the game $\mathcal{G}$ (Prop 5). □

**Remark 1** *The proving idea of approximating by finite games is one feasible and concise way to prove Theorem 1. After analyzing the basic properties in Proposition 1 and 2, the existence of NE can also be proved by the fixed point theorem in (Glicksberg, 1952) while going a little bit of a detour to fit the problem into its proof framework.*

# 5   CCDO & CCDO-RL

In this section, we introduce the Combinatorial-Continuous Double Oracle (CCDO) algorithm and prove its convergence in Section 5.1, and propose the practical version of CCDO, CCDO-RL with the convergence analysis in Sections 5.2 and 5.3 respectively. CCDO has a similar algorithmic framework to DO but differs significantly in the convergence analysis. Moreover, we consider one phenomenon that never occurs in DO but is common in the COPs: the performance of the approximate best response (ABR) to another player's mixed policy is even worse than that of NE. To handle this phenomenon, we design a CCDO approximate (CCDOA) algorithm, and further propose CCDO-RL (Algorithm 1), where we use RL as the underlying oracle solver for both players in the CCDOA framework inspired by recent advancements in using RL to solve COPs. We provide a further convergence analysis on CCDO-RL, and examine how different ABRs influence convergence.

## 5.1   CCDO AND ITS CONVERGENCE

DO is originally proposed to solve NE in the large zero-sum matrix games (McMahan et al., 2003). The key idea is to iteratively compute the mixed NE in the subgame and expand the subgame by the best response (BR) to the current NE of the subgame. We adopt the same algorithmic framework and propose CCDO, to solve the NE in ACCES games (see Algorithm 2). The difference with DO and its variants ODO/XDO is the stopping criterion, Line 6 in Algorithm 2. The original part in DO is subgame sets both remain unexpanded, i.e. $X_{k+1} = X_k, Y_{k+1} = Y_k$ which naturally holds on finite games but cannot be possibly guaranteed in ACCES games, even if iterating infinite times because of $Y$'s infiniteness.

We should not ignore that DO and its variants ODO/XDO can only guarantee convergence under a finite action space because the subgame can become the original game by adding the best response over a finite number of iterations. The infinity of the strategy space not only invalidates this guarantee but also fundamentally alters the structure of convergence analysis. Besides, the two players need to be analyzed separately in the proof because of the asymmetry of ACCES games.

The convergence guarantee of CCDO applies to a broader range of zero-sum games, not only in matrix games but also in the ACCES game. In other words, CCDO is the extensive version of DO. First, we prove the convergence of CCDO, providing the basis for the subsequent convergent proof of CCDOA and CCDO-RL. The full proof is provided in Appendix B.

**Theorem 2** *Given a two-player ACCESS game* $\mathcal{G} = (X, Y, u)$, *where* $X$ *is finite,* $Y$ *is a nonempty compact set, and the utility function* $u$ *is continuous in* $Y$ *when fixing the strategy in* $X$, *we have*

*1. When* $\epsilon = 0$, *every weakly convergent subsequence in the subgame equilibrium sequence* $\{(p_k^*, q_k^*)\}$ *converges to the equilibrium of the whole game, possibly in an infinite number of iterations.*

*2. When $\epsilon > 0$, Algorithm 2 converges to an $\epsilon$- equilibrium in a finite number of epochs.*

## 5.2 CCDOA AND CCDORL

Due to the NP-hardness of most COPs, finding the exact BR for the combinatorial player is computationally impractical. Hence, we use a more practical approximate version, CCDOA (Algorithm 3 of Appendix B), to solve the approximate NE. However, the approximation of BR may cause circumstances where the utility of the approximate best response is lower than that of NE in the subgame which never happens in CCDO. This issue not only has a tricky effect on the convergence analysis but adds computational overhead to solving NE and the memory burden of saving strategies, and may even prolong the iteration round. To address this, two discriminants (lines 6-13 in Algorithm 3) are added to guarantee the optimality of the two ABRs $x_{k+1}^{\epsilon_1}, y_{k+1}^{\epsilon_2}$ in the subgame $\mathcal{G}_k = (X_k, Y_k, u)$.

There have been studies using RL to learn a generalized policy for certain COPs in graphs (Khalil et al., 2017; Nazari et al., 2018; Bengio et al., 2020; Ou et al., 2021; Feng et al., 2025). The key idea is to decompose the node selection into a sequence and learn a heuristic policy for sequentially choosing nodes. The RL policy is usually trained on seen training graphs with the hope of generalizing to unseen test graphs of similar characteristics. Such generalization has been further enhanced via graph embedding techniques such as Structure to Vector (Dai et al., 2016) and Graph Convolutional Networks (Kipf and Welling, 2016) as the underlying value/policy network. The adversary's task is to choose optimal parameters in COPs. RL would also be a useful method to enhance the adversary's generalizability and solvability for diverse instances of the CO problem. Hence, we propose CCDO-RL (see Algorithm 1), a practical implementation of CCDOA where we use RL and graph embedding techniques as the underlying method to find the approximate BR for each player (Line 4 of Algorithm 1). The mixed NE is solved by the supported enumeration algorithm (Roughgarden, 2010), utilizing the Nashpy implementation (Knight and Campbell, 2018).

---
**Algorithm 1** Combinatorial-Continuous Double Oracle Reinforcement Learning Algorithm
---
**Input:** Game $\mathcal{G} = (X, Y, u), \epsilon \geqslant 0$.
**Output:** $\sigma_k^*$.
  1: Initialize strategy set $\Pi_{1,0}, \Pi_{2,0}$.
  2: **repeat**
  3:     Solve the mixed equilibrium $\sigma_k^*$ in the subgame $(\Pi_{1,k}, \Pi_{2,k})$.
  4:     Find the approximate best response by RL algorithms $(\pi_{1,k}^*, \pi_{2,k}^*)$.
  5:     $\Pi_{1,k+1} = \Pi_{1,k} \cup \{\pi_{1,k}^*\}, \Pi_{2,k+1} = \Pi_{2,k} \cup \{\pi_{2,k}^*\}$.
  6:     **if** $U(\pi_{1,k}^*, \sigma_{2,k}^*) \leqslant U(\sigma_k^*)$ **then**
  7:         $\Pi_{1,k+1} = \Pi_{1,k}$.
  8:     **else if** $U(\sigma_{1,k}^*, \pi_{2,k}^*) \geqslant U(\sigma_k^*)$ **then**
  9:         $\Pi_{2,k+1} = \Pi_{2,k}$.
 10:     **end if**
 11: **until** $U(\pi_{1,k}^*, \sigma_{2,k}^*) - U(\sigma_{1,k}^*, \pi_{2,k}^*)) \leqslant \epsilon$
---

## 5.3 CONVERGENCE OF CCDOA AND CCDO-RL

Next, we consider the convergence guarantee of CCDOA and CCDO-RL with ABRs (full proof in Appendix B). Apart from the convergence analysis, the influence of ABRs with different degrees of approximation on the number of algorithm inner iterations is also explained further. Additionally, the computational complexity of CCDO-RL is provided in Theorem 4 in Appendix B.

**Theorem 3** *Given $\mathcal{G} = (X, Y, u)$, where $X$ is finite, $Y$ is a nonempty compact set, and the utility function $u$ is continuous in $Y$ when fixing the strategy in $X$, with $\epsilon_1$ best response oracle for Player 1 in $X$ and $\epsilon_2$ best response oracle for Player 2 in $Y$, we have*

*1. When $\epsilon > 0$, for any form of approximate best response oracles, CCDOA and CCDO-RL can converge to a finitely supported $(\epsilon + \epsilon_1 + \epsilon_2)$- equilibrium in a finite number of iterations.*

*2. When $\epsilon = 0$, if the approximate response oracle for Player 2 has a uniform lower bound for every mixed strategy in $\triangle_X$, i.e.*

$$\forall p \in \triangle_X, \exists \epsilon_Y, s.t. U(p, BR_2^{\epsilon_2}(p)) \geqslant \min_{y \in Y} U(p, y) + \epsilon_Y, \tag{6}$$

*then CCDOA and CCDO-RL must converge to an $(\epsilon + \epsilon_1 + \epsilon_2)$- equilibrium in a finite iterations.*

*3. When $\epsilon = 0$, if CCDOA and CCDO-RL produce infinite iterations, every weakly convergent subsequence converges to an $\epsilon_1$- equilibrium.*

The convergence result for $\epsilon > 0$ in Theorem 3 Item 1 is similar to Theorem 2 Item 2, converging to the approximate NE in a finite number of iterations. If the iteration continues indefinitely, the approximation of NE found by CCDO-RL depends solely on that of Player 1's ABR, i.e. $\epsilon_1$. When $\epsilon = 0$, CCDO may continue for an infinite number of iterations in the same problem setting. In contrast, CCDO-RL can terminate in finite rounds if the approximate error of ABRs for Player 2 is bounded below by $\epsilon_Y$ or the condition in Remark 2 is satisfied by Player 1 or 2.

**Remark 2** *Except for the uniform lower bound for Player 2 in (6), if two absolute differences between BR and ABR do not converge to zero, including divergence and convergence to a positive number, i.e.*

$$\max_{x \in X} U(x, q_k^*) - U(x_{k+1}^{\epsilon_1}, q_k^*) \nrightarrow 0 \quad or \quad U(p_k^*, y_{k+1}^{\epsilon_2}) - \max_{y \in Y} U(p_k^*, y) \nrightarrow 0,$$

*then CCDOA and CCDO-RL must terminate in a finite number of iterations, even if $\epsilon = 0$*

## 6 EXPERIMENTS

With theoretical guarantees on the existence and convergence of NE for ACCES games, we are also interested in how our proposed algorithm CCDO-RL works empirically. To evaluate this, we conduct experiments of CCDO-RL on three distinct ACCES game instances introduced in Section 6.1 and analyze the performance of CCDO-RL in Section 6.2. Section 6.2.1 aims to empirically demonstrate the convergence (Figures 1 and 2) of the algorithm CCDO-RL over realistic CO problems, and show its consistency with Theorem 3. Section 6.2.2 intends to show the average reward (to seen training graphs) as well as the generalizability (to unseen test graphs) of the combinatorial player in real-world ACCES games (shown in Tables 1, and 2).

### 6.1 THREE INSTANCES OF ACCES GAMES

We consider a certain COP which is parameterized with $\{\theta_i\}$, where $i$ is the index of nodes (such as a target in security games) – e.g., such parameters can be interpreted as attack probability of targets. In real-world applications, we often need to estimate such parameters before solving the COPs. Unfortunately, the estimation $\{\hat{\theta}_i\}$ often bears a gap to the true value $\{\theta_i\}$, which derives from e.g. environment (aleatoric) uncertainty, model (epistemic) uncertainty, or an attacker trying to manipulate the defender's utility. We use a generic model to formulate this gap:

$$\theta_i = \hat{\theta}_i + y \cdot \tau_i, \tag{7}$$

where $y$ represents the strategy of the nature/attacker, $\tau_i$ is the environment factors like weather and transportation conditions, or human subjective factors like the preference of the attacker. Such abstraction can represent a wide range of ACCES games, such as facility location covering problems An (2020); Tirkolaee et al. (2020), CVRP Gendreau et al. (2014); Dinh et al. (2018); Florio et al. (2023), security patrolling (OP) (Xu et al., 2021), and influence maximization problem Kalimeris et al. (2019). We describe three instances of ACCES games based on the model (7).

**Adversarial Covering Salesman Problem (ACSP):** In a map of cities, every city $i$ has a coverage $\theta_i$. A salesman finds the shortest path such that all cities are visited or covered, with $\theta_i$ influenced by physical factors $\tau_i$ and transportation parameters $y$ based on Eq.(7). The salesman is Player 1 where $X$ consists of the feasible paths of the salesman. Nature is Player 2 with $Y = [0, 1]^K \ni y, K \in \mathbb{N}$. The utility function of Player 1 $u$ is the opposite of the total traveling distance.

**Adversarial Capacitated Vehicle Routing Problem (ACVRP):** A vehicle with a constrained capacity of goods finds the shortest path under the worst case with the $i_{th}$ customer's demand $\theta_i$ changed by environmental factors $\tau_i$ and weather parameter $y$ on Eq.(7). The vehicle is Player 1 where $X$ is the set of the feasible path $x$. Nature is Player 2 where $Y$ is $[0, 1]^K \ni y, K \in \mathbb{N}$. The utility function of Player 1 $u$ is the opposite of total delivery distance satisfying all the demands of customers.

**Patrolling Game (PG):** The patrolling game is described in the introduction.

For all the problem instances, we run our algorithm on two problem sizes: 20 nodes and 50 nodes. The detailed description and problem parameters of the three game instances are in Appendix D.

## 6.2 PERFORMANCE OF CCDO-RL

Two aspects are evaluated for the performance of CCDO-RL, i.e., i) Convergence to NE (Section 6.2.1) exploring whether CCDO-RL can compute the NE, and ii) Protagonist policy's average reward and generalizability (Section 6.2.2). Generalizability refers to the ability of RL models trained on previously seen graphs (problem instances), to perform well on a new set of unseen test graphs. The model's usability is enhanced by generalizability, rather than focusing solely on the average reward, which is a critical motivation in the literature on RL for COPs (Khalil et al., 2017; Kool et al., 2018).

For all the problems, CCDO-RL adopts the REINFORCE algorithm with an attention-based encoder-decoder framework (Kool et al., 2018) (used as an inductive graph representation component) to learn a generalizable COP solver for Player 1 (protagonist), and PPO to train a policy for Player 2 (adversary) whose strategy space is continuous. CCDO-RL is trained on a set of 10,000 graphs (with 20 or 50 nodes). The hyperparameters of CCDO-RL are specified in Appendix D (Table 3). Our code is included as supplementary material and will be open-sourced for ease of reproduction.

### 6.2.1 CONVERGENCE TO NE

Exploitability is a common metric to describe the closeness to true NE by calculating the sum of performance distances between each new best response and subgame NE, i.e. $\sum_{i=1,2} U(\pi_{i,k}^{br}, \sigma_{-i,k}) - U(\sigma)$ in the general two-player game. Since our game is zero-sum, the calculation is as follows:

$$\text{Exploitability}(\sigma) = \max_{\pi_1 \in \Sigma_1} U(\pi_1, \sigma_2) - \min_{\pi_2 \in \Sigma_2} U(\sigma_1, \pi_2).$$

From Figure 1, we can see that CCDO-RL can converge to approximate NE in 25 iterations or less (in the PG setting), reaching 0.05 in ACSP, 0.10 in ACVRP, and 0.03 in PG with 20 nodes. Similar results are observed in problems with 50 nodes (see Figure 2 in Appendix F). These results validate the effectiveness of CCDO-RL in finding the NE for various types of games.

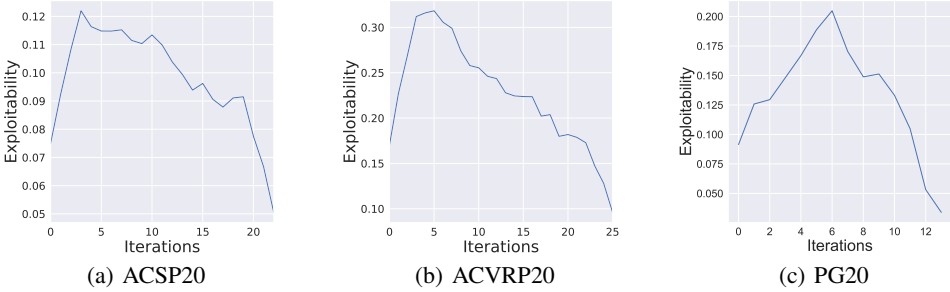

(a) ACSP20        (b) ACVRP20        (c) PG20

Figure 1: Exploitability curve of CCDO-RL on three games of 20 nodes

### 6.2.2 AVERAGE REWARD AND GENERALIZABILITY OF COMBINATORIAL PLAYER

**Evaluation.** The learned policies are tested on 200 graphs, with 100 being randomly selected from the 10,000 training graphs (to show the average reward), and the other 100 being unseen graphs (to test policy generalization). We evaluate the performance of the protagonist with the adversary under three COPs. For each COP, the performance is considered both on the 20-node and 50-node map.

**Baselines.** There are heuristic algorithms for each game instance (Heuristic in Table 1 and 2) and a single-player RL algorithm. For ACVRP, we adopt the Tabu Search algorithm (Tabu) (Li and Li, 2020) as the heuristic algorithm, which is widely applied in the routing problem. For ACSP, the common benchmark local search algorithm, LS2 (Golden et al., 2012), is used. For PG, we choose the greedy algorithm as the baseline. The "RL against Stoc" algorithm in Tables 1 and 2 is identical to the protagonist model in CCDO-RL but trained in environments with stochastic adversarial perturbations.

**Average Reward.** As illustrated in Table 1, our algorithm achieves a better average reward than baselines (10.08% improvement on average of all settings against two baselines), regardless of CO instance or problem size, when confronting the adversary trained by CCDO-RL. In the setting of CSP-20 nodes, the average reward is improved by 46.98% compared to the heuristic and by 7.14% compared with the RL against Stoc. For the 50-node setting, the improvements are 45.91% and 5.28% respectively. Similarly, the improvements in contrast to Heuristic and RL against Stoc are as follows: 1.72% and 3.01% for CVRP-20 nodes, 0.75% and 4.46% for CVRP-50 nodes, 4.17% and 1.48% for PG-20 nodes, and 10.60% and 4.38% for PG-50 nodes.

**Generalizability.** From Table 2, CCDO-RL continues to achieve a better average reward when facing the adversary, demonstrating that the learned RL policies generalize well to unseen graphs. Even though the non-RL baselines do have access to the graph structures and other problem information of the unseen problem instances, CCDO-RL can obtain comparable performances without re-training on the new problem instances. The improvements versus Heuristic and RL against Stoc are 46.61% and 7.02% for CSP-20 nodes, 42.24% and 3.94% for CSP-50 nodes, 1.12% and 1.56% for CVRP-20 nodes, 0.90% and 5.05% for CVRP-50 nodes, 5.35% and 2.40% for PG-20 nodes, and 12.17% and 10.33% for PG-50 nodes. Even when confronting the stochastic adversary, CCDO shows superior generalizability compared to two baselines across three COPs, with average improvements of 6.31%, 3.42%, and 3.95% respectively. Detailed results are provided in Appendix F (Tables 5 - 10).

**Remark 3** *In CO problems (or more broadly, operations research and economics), it is known that achieving solution quality improvements against strong baselines (e.g., the RL methods trained with a stochastic adversary) is very challenging, and the margins are usually small (Kool et al., 2018), sometimes even less than 1%. However, these "tiny" marginal improvements in profits keep small business owners in the real world alive. Last, the improvement depends a lot on the problem settings, and we show that sometimes the improvement can be much more significant.*

Table 1: Average reward against CCDO-RL's adversary (on seen graphs)

| method | ACSP (Mean±Std) | | ACVRP (Mean±Std) | | PG (Mean±Std) | |
|---|---|---|---|---|---|---|
| | 20 nodes | 50 nodes | 20 nodes | 50 nodes | 20 nodes | 50 nodes |
| Heuristic | 6.13±1.20 | 7.55±1.42 | 7.65±1.23 | 13.38±1.70 | 2.64±1.03 | 4.53±1.84 |
| RL against Stoc | 3.50±0.47 | 4.55±0.62 | 7.55±1.16 | 13.90±1.63 | 2.71±0.90 | 4.80±2.18 |
| CCDO-RL | **3.25**±0.42 | **4.31**±0.51 | **7.42**±1.21 | **13.28**±1.52 | **2.75**±0.87 | **5.01**±1.91 |

Table 2: Generalizability against CCDO-RL's adversary (on unseen graphs)

| method | ACSP (Mean±Std) | | ACVRP (Mean±Std) | | PG (Mean±Std) | |
|---|---|---|---|---|---|---|
| | 20 nodes | 50 nodes | 20 nodes | 50 nodes | 20 nodes | 50 nodes |
| Heuristic | 6.20±1.33 | 7.60±1.37 | 7.64±1.30 | 13.27±1.87 | 2.43±0.98 | 4.19±1.69 |
| RL against Stoc | 3.56±0.37 | 4.57±0.58 | 7.67±1.30 | 13.85±1.53 | 2.50±0.95 | 4.26±2.17 |
| CCDO-RL | **3.31**±0.35 | **4.39**±0.52 | **7.55**±1.28 | **13.15**±1.59 | **2.56**±0.92 | **4.70**±1.94 |

[1] For the average reward of ACSP and ACVRP, smaller is better while for that of PG larger is better.

## 7    CONCLUSION & LIMITATIONS

Drawing insights from existing literature and real-world applications, we define a new class of games called ACCES games. We prove the existence of NE for ACCES games, providing a fundamental basis for solution algorithms. Two NE solvers are introduced, namely CCDO and its practical version CCDO-RL, along with original theoretical analysis and ABRs' impact on convergence. Empirical results show that CCDO-RL can converge to approximate NE in a small number of iterations. The protagonist policy obtained via CCDO-RL has better average rewards against adversarial perturbations and shows great generalizability on unseen graphs. A potential limitation of our method is scalability – our experiments mainly focus on small COPs (20 and 50 nodes). While scalability is not the focus of this study, it does remain unexplored and deserves more investigation. Our work also opens up a new area of research centering on ACCES games, and more broadly asymmetric games such as the uniqueness of NE, as well as more efficient and practical algorithms.

## 8 ACKNOWLEDGMENTS

This research was supported by the National Science Foundation under IIS-2348405. The authors acknowledge William & Mary Research Computing for providing computational resources and/or technical support that have contributed to the results reported within this paper.

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

## A    PROOFS AND ANALYSIS IN SECTION 4

### A.1    PROOFS IN SECTION 4

**Definition 3** [Weakly Convergence.] *Suppose $S$ is a space, probability measures $P_n$ weakly converges to $P$, written by $P_n \Rightarrow P$, if for every bounded continuous function $f$,*

$$\lim_{n \to \infty} \int_S f dP_n \to \int_S f dP.$$

If any sequence in a set has a weakly convergent subsequence, the set is weakly sequentially compact.

**Lemma 1** *$\mathcal{S}', \mathcal{S}''$ are uncorrelated general metric spaces and $P', P''$ are the probability measure on $\mathcal{S}', \mathcal{S}''$ respectively. Define $\mathcal{T} \triangleq \mathcal{S}' \times \mathcal{S}''$ as the product space of $\mathcal{S}'$ and $\mathcal{S}''$. if $\mathcal{T}$ is separable, then $P'_n \times P''_n \Rightarrow P' \times P''$ if and only if $P'_n \Rightarrow P'$ and $P''_n \Rightarrow P''$. ((Myerson, 1991), Theorem 2.8)*

**Proposition 1** [Weakly Sequential Compactness.] Suppose the ACCES game is $\mathcal{G} = (X, Y, u)$, where $X$ is finite, $Y$ is a nonempty compact metric space, and the utility function $u$ is continuous on $Y$ fixing $x \in X$. Then the joint mixed strategy space $\triangle \triangleq \triangle_X \times \triangle_Y$ is weakly sequentially compact.

*Proof.*    Firstly, considering the condition in Lemma 1, we need to prove the product space $X \times Y$ is separable. The set $X$ is separable obviously, based on its finiteness and discreteness. And every compact metric space has a countable base, so separable. Hence the set $Y$ is separable. Then the product space $X \times Y$ is separable too.

The next step is to prove weakly sequential compactness of set $\triangle_X$ and $\triangle_Y$. Due to the finiteness of $X$, $\triangle_X$ is a nonempty compact convex set on $\mathbb{R}^{|X|}$ where any element $p \in \triangle_X$ can be represented as $p = [p(x_1), ..., p(x_{|X|})]$ satisfying $\sum_{i=1}^{|X|} p(x_i) = 1, p_i \geqslant 0$. Since strong convergence is equivalent to weak convergence in the finite-dimensional normed space, the compact set $\triangle_X$ is weakly sequentially compact too.

Besides, according to the properties of mixed strategies in continuous games mentioned in Liu et al. (2007), $\triangle_Y$ is sequentially compact and closed, thus compact. Based on its compactness, proposition 1 of Adam et al. (2021) guarantees that $\triangle_Y$ is weakly sequentially compact.

Therefore, due to Lemma 1, we can get $\triangle$ is weakly sequentially compact.    □

**Proposition 2** [Continuity of Expected Utility Function.] The expected utility function $U(p, q) \triangleq \sum_{x \in X} \int_{y \in Y} p(x) u(x, y) dq$ is continuous on the joint mixed strategy space $\triangle$, $\forall p \in \triangle_X, q \in \triangle_Y$.

*Proof.*    First we denote the related distance mapping $\rho_1$ and $\rho_2$ on $\triangle_X$ and $\triangle_Y$ respectively.

$$\rho_1(p, p') = \sum_{x \in X} |p(x) - p'(x)|, \forall p, p' \in \triangle_X,$$

$$\rho_2(q, q') = \sup_{y \in Y} |q(y) - q'(y)|, \forall q, q' \in \triangle_Y.$$

Afterwards, we prove the continuousness of $U$ on $\triangle$. $\forall (p_0, q_0) \in \triangle, \forall (p, q) \in O((p_0, q_0), \delta) \cap \triangle$, which means $d((p, q), (p_0, q_0)) \triangleq \sqrt{\rho_1^2(p, p_0) + \rho_2^2(q, q_0)} \leqslant \delta$,

$$|U(p, q) - U(p_0, q_0)| = |\sum_{x \in X} p(x) \int_{y \in Y} u(x, y) dq - \sum_{x \in X} p_0(x) \int_{y \in Y} u(x, y) dq_0| \tag{1}$$

$$= |\sum_{x \in X} \int_{y \in Y} u(x, y)(p(x) dq - p_0(x) dq_0)| \tag{2}$$

$$= |\sum_{x \in X} \int_{y \in Y} u(x, y)[(p(x) - p_0(x)) dq + p_0(x)(dq - dq_0)]| \tag{3}$$

$$\leqslant |\sum_{x \in X} \int_{y \in Y} u(x, y)(p(x) - p_0(x)) dq| + |\sum_{x \in X} \int_{y \in Y} u(x, y) p_0(x)(dq_0 - dq)| \tag{4}$$

Because $u(x, y)$ is continuous on $Y$ when fixing $x \in X$, $X$ is finite, and $Y$ is nonempty compact metric space, $u(x, y)$ is bounded.

Assume $|u(x, y)| \leqslant M, d((p, q), (p_0, q_0)) \triangleq \sqrt{\rho_1^2(p, p_0) + \rho_2^2(q, q_0)} \leqslant \delta = \frac{\epsilon}{2M}$, we can get that

$$|\sum_{x \in X} \int_{y \in Y} u(x, y)(p(x) - p_0(x)) dq| \leqslant |\sum_{x \in X} (p(x) - p_0(x)) \int_{y \in Y} u(x, y) dq| \tag{5}$$

$$\leqslant M |\sum_{x \in X} (p(x) - p_0(x))| \leqslant M \rho_1(p, p_0) \tag{6}$$

$$|\sum_{x \in X} \int_{y \in Y} u(x, y) p_0(x)(dq_0 - dq)| \leqslant |\sum_{x \in X} p_0(x) \int_{y \in Y} u(x, y)(dq_0 - dq)| \tag{7}$$

$$\leqslant \sum_{x \in X} p_0(x) |\int_{y \in Y} u(x, y)(dq_0 - dq)| \tag{8}$$

$$\leqslant \sup_{x \in X} |\int_{y \in Y} u(x, y)(dq_0 - dq)| \tag{9}$$

$$\leqslant M \rho_2(q, q_0) \tag{10}$$

Then we can get that

$$|U(p, q) - U(p_0, q_0)| \leqslant M \rho_1(p, p_0) + \sum_{x \in X} p_0(x) |\int_{y \in Y} u(x, y) dq - \int_{y \in Y} u(x, y) dq_0| \tag{11}$$

$$\leqslant M\delta + M\delta = 2M\delta = \epsilon. \tag{12}$$

When $p_n \Rightarrow p$ and $q_n \Rightarrow q$, the above inequality still holds. $p_n \Rightarrow p \Leftrightarrow \rho_1(p_n, p) \to 0$ because Strong convergence is equivalent to weak convergence in finite-dimension metric spaces. Additionally, $|\int_{y \in Y} u(x, y) dq - \int_{y \in Y} u(x, y) dq_n| \to 0$ when fixing $x$. From inequality (11), we can infer

$$|U(p, q) - U(p_n, q_n)| \leqslant M \rho_1(p, p_n) + \sum_{x \in X} p_n(x) |\int_{y \in Y} u(x, y) dq - \int_{y \in Y} u(x, y) dq_n| \tag{13}$$

$$\leqslant M \rho_1(p, p_n) + \sup_{x \in X} |\int_{y \in Y} u(x, y) dq - \int_{y \in Y} u(x, y) dq_n| \tag{14}$$

$$= M \rho_1(p, p_n) + |\int_{y \in Y} u(\hat{x}, y) dq - \int_{y \in Y} u(\hat{x}, y) dq_n| \to 0, \tag{15}$$

where $\hat{x} = argmax_{x \in X} |\int_{y \in Y} u(\hat{x}, y) dq - \int_{y \in Y} u(\hat{x}, y) dq_n|$. □

**Proposition 3** [Approximate NE of ACCES.] $\mathcal{G}' = \langle X, Y, \tilde{u} \rangle$ is $\alpha$-approximation of $\mathcal{G} = \langle X, Y, u \rangle$, where $\mathcal{G}$ is an ACCES game. $(p^*, q^*)$ is an $\epsilon$-equilibrium of $\mathcal{G}'$, then $(p^*, q^*)$ is an $(\epsilon + 2\alpha)$-equilibrium of $\mathcal{G}$.

*Proof.* Define $\tilde{U}(p, q) = \sum_{x \in X} \int_{y \in Y} p(x)\tilde{u}(x, y)dq$, for $\forall (p, q) \in \triangle$ we can get

$$|U(p, q) - \tilde{U}(p, q)| = |\sum_{x \in X} \int_{y \in Y} p(x)(u(x, y) - \tilde{u}(x, y))dq| \leqslant \alpha \sum_{x \in X} \int_{y \in Y} p(x)dq = \alpha$$

Next for any $q \in \triangle_Y$,

$$|U(p^*, q^*) - U(p^*, q)| = |U(p^*, q^*) - \tilde{U}(p^*, q^*) + \tilde{U}(p^*, q^*) - \tilde{U}(p^*, q) + \tilde{U}(p^*, q) - U(p^*, q)| \leqslant 2\alpha + \epsilon.$$

Similarly, it can be proved that $|U(p^*, q^*) - U(p, q^*)| \leqslant 2\alpha + \epsilon, \forall p \in \triangle_X$. □

**Proposition 4** [Essentially Finite of ACCES.] For an ACCES $\mathcal{G}$, $\forall \alpha > 0$, there exists an essentially finite strategic game $\hat{\mathcal{G}} = \langle X, \hat{Y}, \hat{u} \rangle$, s.t. $\hat{\mathcal{G}}$ is $\alpha$-approximation of $\mathcal{G}$.

*Proof.* Due to that $\hat{\mathcal{G}}$ is $\alpha$-approximation of $\mathcal{G}$, then for any $x \in X, y \in Y$, $|u(x, y) - \hat{u}(x, y)| \leqslant \alpha$. On account that $u$ is continuous on $Y$ and $Y$ is a nonempty compact metric space, so $u$ is uniformly continuous on $Y$. According to the uniform continuousness of function $u$, $\forall \alpha > 0, \exists \epsilon(x) > 0$, when $|y - y'| < \epsilon(x)$, $|u(x, y) - u(x, y')| \leqslant \alpha$. Define $\epsilon = \min_{x \in X} \epsilon(x)$. Y is a nonempty compact metric space, hence it can be covered by finite open balls $O_j(y_j, \epsilon)$, i.e. $Y \subset \bigcup_j O_j(y_j, \epsilon)$.

Then for $\forall j$, $\forall y \in O_j(y_j, \epsilon), x \in X$, denote that $\hat{u}(x, y) = u(x, y_j)$. Hence, $\forall x \in X, \forall y \in Y$, $|u(x, y) - \hat{u}(x, y)| \leqslant \alpha$. □

**Theorem 1** [Existence of NE] $\mathcal{G} = \langle X, Y, u \rangle$, *where $X$ is finite space, $Y$ is nonempty compact metric space, $u : X \times Y \to \mathbb{R}$ is a continuous utility function on $Y$ when fixing $x \in X$. Game $\mathcal{G}$ has a mixed strategy Nash equilibrium.*

*Proof.* Suppose sequence $\{\alpha_n\}$ converges to zero, i.e. $\alpha_n \to 0$. For any $\alpha_n$, there exists an essentially finite game $\mathcal{G}_n$, which is $\alpha_n$-approximation of $\mathcal{G}$ (Proposition 4). Due to Nash's theorem, mixed equilibrium $(p_n, q_n)$ of $\mathcal{G}_n$ exists. So $(p_n, q_n)$ is $2\alpha_n$-equilibrium of $\mathcal{G}$ (Proposition 3). By Proposition 1, $(p_n, q_n)$ has a convergent subsequence. For brevity, this convergent subsequence is denoted by $\{(p_n, q_n)\}$, which converges to $(p^*, q^*)$. Based on Proposition 5, we can know that $(p^*, q^*)$ is a mixed equilibrium of $\mathcal{G}$. □

## A.2 ANALYSIS ON THE EXISTENCE OF NE IN N-PLAYER ACCES GAMES

Our propositions and Theorem 2 can be extended to the N-player ACCES games naturally. The key point of the existence of NE to N-player ACCES games is two fundamental properties we propose in ACCES games, weakly sequential compactness of the mixed strategy space and continuity of the expected utility function (Propositions 1 and 2), and the approximation idea by finite games. We introduce these as follows.

- **Two Properties**: In Proposition 1, we transform the weakly sequential compactness of the joint mixed strategy space into the separability and weakly sequential compactness of each single player by Lemma 1. In Proposition 2, we scale the distance between two mixed strategies to the sum of distances between a single player's mixed strategies while fixing other players. According to the proof of these two propositions, they are all independent of the number of players.

- **The Approximation idea by finite games**: The main idea is to approximate the infinite continuous strategy space by finite grids by definitions of approximate games and essentially finite games. The idea and definitions are not limited to the two-player setting.

## B  PROOFS IN SECTION 5

**Theorem 2** *Given a two-player ACCESS game $\mathcal{G} = (X, Y, u)$, where $X$ is finite, $Y$ is a nonempty compact set, and the utility function $u$ is continuous in $Y$ when fixing the strategy in $X$, we have*

*1. When $\epsilon = 0$, every weakly convergent subsequence in the subgame equilibrium pair sequence $\{(p_k^*, q_k^*)\}$ converges to the equilibrium of the whole game, possibly in an infinite number of iterations.*

*2. When $\epsilon > 0$, Algorithm 2 converges to an $\epsilon$- equilibrium in a finite number of epochs.*

*Proof.*    At every epoch $k$, denote the protagonist policy $x_{k+1} \in X$ and adversary policy output $y_{k+1} \in Y$ as best responses to mixed equilibrium $(p_k^*, q_k^*)$ in the $k_{th}$ subgame $(X_k, Y_k, U)$, i.e.

$$x_{k+1} = \mathrm{argmax}_{x \in X} U(x, q_k^*), y_{k+1} = \mathrm{argmin}_{y \in Y} U(p_k^*, y),$$

noting that all maximizers and minimizers exist due to the finiteness of $X$, compactness of $Y$, and continuity of $u$ when fixing variable $x$.

First, prove the efficiency of the stopping criterion, i.e. output $(p_k^*, q_k^*)$, satisfying this criterion, must be $\epsilon$- mixed equilibrium of game $\mathcal{G}$. The stopping criterion

$$U(x_{k+1}, q_k^*) - U(p_k^*, y_{k+1}) \leqslant \epsilon,$$

implies that

$$
\begin{aligned}
U(p_k^*, q_k^*) \leqslant U(x_{k+1}, q_k^*) &\leqslant U(p_k^*, y_{k+1}) + \epsilon \\
&= \min_{y \in Y} U(p_k^*, y) + \epsilon = \min_{q \in \triangle_Y} U(p_k^*, q) + \epsilon,
\end{aligned}
\tag{16}
$$

which means that $\forall q \in \triangle_Y, U(p_k, q_k) \leqslant U(p_k, q) + \epsilon$.

Similarly, we can get that

$$
\begin{aligned}
U(p_k^*, q_k^*) \geqslant U(p_k^*, y_{k+1}) &\geqslant U(x_{k+1}, q_k^*) - \epsilon \\
&= \max_{x \in X} U(x, q_k^*) - \epsilon = \max_{p \in \triangle_X} U(p, q_k^*) - \epsilon,
\end{aligned}
\tag{17}
$$

that is $\forall p \in \triangle_X, U(p_k^*, q_k^*) \geqslant u(p, q_k^*) - \epsilon$. Combined (16) and (17), mixed equilibrium $(p_k^*, q_k^*)$, meeting the condition, is a $\epsilon$-mixed equilibrium.

Next, we need to prove its convergence of mixed Nash equilibrium, in other words, this algorithm 2 can reach the terminal condition.

When $\epsilon = 0$, due to Proposition 1, every sequence in $\triangle_{X \times Y}$ has its own weakly convergent subsequence. For the sequence $\{(p_k^*, q_k^*)\}$, whose element $(p_k^*, q_k^*)$ is the mixed equilibrium of the $k_{th}$ subgame, there exists a weakly convergent subsequence, for simplicity denoted the same indices, i.e. $\{(p_k^*, q_k^*)\}$ converges to $(p^*, q^*)$.

Due to that $(p_k^*, q_k^*)$ is an equilibrium of the subgame $(X_k, Y_k, U)$, so $\forall k, \forall y \in Y_k$, we can know that

$$U(p_k^*, q_k^*) \leqslant U(p_k^*, y).$$

Take the limit on both sides of this inequality, based on the continuousness of U on $\triangle_{X \times Y}$, we can get that

$$U(p^*, q^*) \leqslant U(p^*, y), \forall y \in cl(\cup Y_k). \tag{18}$$

Because $Y$ is compact, so for sequence $\{y_k\}$ there exists $\hat{y} \in cl(\cup Y_k)$ s.t. $y_k \to \hat{y}, k \to \infty$.

Besides, we can get that

$$U(p_k^*, y_{k+1}) \leqslant U(p_k^*, y), \forall y \in Y, \tag{19}$$

based on the definition of best response, which can infer

$$U(p^*, q^*) \leqslant U(p^*, \hat{y}) \leqslant U(p^*, y), \forall y \in Y, \tag{20}$$

in which the left-hand inequality above follows inequality (18) and the right-hand gets by taking limits on inequality (19).

For the reason that the strategy space $X$ is finite and the number of iterations is infinite, in the weakly convergent subsequence,

$$\exists k_0, s.t. \forall k > k_0, U(p_k^*, q_k^*) = \max_{x \in X_k} U(x, q_k^*)$$
$$= \max_{x \in X} U(x, q_k^*) = U(x_{k+1}, q_k^*), \tag{21}$$

which means that $x_{k+1} \in X_k, \forall k > k_0$. Therefore we can get that, $\forall x \in X$,

$$U(x, q_k^*) \leqslant U(x_{k+1}, q_k^*) = U(p_k^*, q_k^*), \forall k > k_0,$$
$$\implies U(x, q^*) \leqslant U(p^*, q^*). \tag{22}$$

So we have proven that

$$U(x, q^*) \leqslant U(p^*, q^*) \leqslant U(p^*, y), \forall x \in X, \forall y \in Y.$$

Due to an equivalent condition to NE, that is $(p^*, q^*)$ is an equilibrium if and only if it follows

$$U(x, q^*) \leqslant U(p^*, q^*) \leqslant U(p^*, y), \forall x \in X, y \in Y, \tag{23}$$

we can says that $(p^*, q^*)$ is an equilibrium of $\mathcal{G}$.

If $\epsilon > 0$, (20) imply that

$$U(p_k^*, y_{k+1}) \leqslant U(p_k^*, q_k^*) \implies U(p^*, \hat{y}) \leqslant U(p^*, q^*). \tag{24}$$

Combined with inequality (20), we know that

$$U(p_k^*, y_{k+1}) \to U(p^*, \hat{y}) = U(p^*, q^*). \tag{25}$$

Due to the fact (21), after $k_0$ iterations, the strategy space of $X_k$ will not be expanded. So we can get

$$\lim_{k \to \infty} U(x_{k+1}, q_k^*) = \lim_{k \to \infty} U(p_k^*, q_k^*) = U(p^*, q^*). \tag{26}$$

Therefore, utilizing (25) and (26),

$$U(x_{k+1}, q_k^*) - U(p_k^*, y_{k+1}) \to 0, k \to \infty. \tag{27}$$

In other words, if $\epsilon > 0$, this iterated process must be terminated within limited rounds and the output is $\epsilon$- equilibrium with finite supports. □

**Theorem 3** *Given $\mathcal{G} = (X, Y, u)$, where $X$ is finite, $Y$ is a nonempty compact set, and the utility function $u$ is continuous in $Y$ when fixing the strategy in $X$, with $\epsilon_1$ best response oracle for Player 1 in $X$ and $\epsilon_2$ best response oracle for Player 2 in $Y$, we have*

*1. When $\epsilon > 0$, for any form of approximate best-response oracles, CCDOA and CCDO-RL can converge to a finitely supported $(\epsilon + \epsilon_1 + \epsilon_2)$- equilibrium in a finite number of iterations.*

*2. When $\epsilon = 0$, if the approximate response oracle for Player 2 has a uniform lower bound for every mixed strategy in $\triangle_X$, i.e.*

$$\forall p \in \triangle_X, \exists \epsilon_Y, s.t. U(p, BR_2^{\epsilon_2}(p)) \geqslant \min_{y \in Y} U(p, y) + \epsilon_Y, \tag{28}$$

*then CCDOA and CCDO-RL must converge to an $(\epsilon + \epsilon_1 + \epsilon_2)$- equilibrium in a finite iterations.*

*3. When $\epsilon = 0$, if CCDOA and CCDO-RL produce infinite iterations, every weakly convergent subsequence converges to an $\epsilon_1$- equilibrium.*

*Proof.* (1) Due to that

$$U(x_{k+1}^{\epsilon_1}, q_k^*) - U(p_k^*, y_{k+1}^{\epsilon_2}) \leqslant U(x_{k+1}, q_k^*) - U(p_k^*, y_{k+1}),$$

combined with (27), the Algorithm 3 can stop in a finite number of iterations if $\epsilon > 0$.

(2) Similarly with Theorem 2, firstly we prove the output must be $(\epsilon + \epsilon_1 + \epsilon_2)$- equilibrium if satisfying the stopping termination. Based on the definition of $\epsilon$- best response, we can know that

$$\begin{aligned} \forall q \in \triangle_Y, U(BR_1^{\epsilon_1}(q), q) \geqslant \max_{x \in X} U(x, q) - \epsilon_1, \\ \forall p \in \triangle_X, U(p, BR_2^{\epsilon_2}(p)) \leqslant \min_{y \in Y} U(p, y) + \epsilon_2. \end{aligned} \tag{29}$$

For simplicity, suppose $BR_1^{\epsilon_1}(q_k^*) \triangleq x_{k+1}^{\epsilon_1}, BR_2^{\epsilon_2}(p_k^*) \triangleq y_{k+1}^{\epsilon_2}$ for subgame equilibrium $(p_k^*, q_k^*)$. The iteration process stops means that $U(x_{k+1}^{\epsilon_1}, q_k^*) - U(p_k^*, y_{k+1}^{\epsilon_2}) \leqslant \epsilon$. So we can get that

$$\begin{aligned} U(p_k^*, q_k^*) \leqslant \max_{x \in X} U(x, q_k^*) &\leqslant U(x_{k+1}^{\epsilon_1}, q_k^*) + \epsilon_1 \\ &\leqslant U(p_k^*, y_{k+1}^{\epsilon_2}) + \epsilon + \epsilon_1 \\ &\leqslant \min_{y \in Y} U(p_k^*, y) + \epsilon + \epsilon_1 + \epsilon_2 \\ &\leqslant U(p_k^*, y) + \epsilon + \epsilon_1 + \epsilon_2, \forall y \in Y. \end{aligned} \tag{30}$$

Similarly, we can prove the parallel results on $X$.

$$\begin{aligned} U(p_k^*, q_k^*) \geqslant \min_{y \in Y} U(p_k^*, y) &\geqslant U(p_k^*, y_{k+1}^{\epsilon_2}) - \epsilon_2 \\ &\geqslant U(x_{k+1}^{\epsilon_1}, q_k^*) - \epsilon - \epsilon_1 \\ &\geqslant \max_{x \in X} U(x, q_k^*) - \epsilon - \epsilon_1 - \epsilon_2 \\ &\geqslant U(x, q_k^*) - \epsilon - \epsilon_1 - \epsilon_2, \forall x \in X. \end{aligned} \tag{31}$$

So combined with (30) and (31),

$$\forall x \in X, \forall y \in Y, U(x, q_k^*) - \bar{\epsilon} \leqslant U(p_k^*, q_k^*) \leqslant U(p_k^*, y) + \bar{\epsilon},$$

in which $\bar{\epsilon} = \epsilon + \epsilon_1 + \epsilon_2$. Hence $(p_k^*, q_k^*)$ is the $\bar{\epsilon}$- equilibrium of game $\mathcal{G}$. If the $\epsilon_2$- best response has a lower bound, which means that $\forall i$,

$$U(p_k^*, y_{k+1}) + \epsilon_Y \leqslant U(p_k^*, y_{k+1}^{\epsilon_2}) \leqslant U(p_k^*, q_k^*), \tag{32}$$

assume $y_{k+1}^{\epsilon_2} \to \hat{y}^{\epsilon_2}, k \to \infty$, if iterating infinitely, take limits on both sides, based on (25) we can get

$$U(p^*, q^*) + \epsilon_Y = U(p^*, \hat{y}) + \epsilon_Y \leqslant U(p^*, \hat{y}^{\epsilon_2}) \leqslant U(p^*, q^*). \tag{33}$$

Obviously, it's a contradiction. Hence this iterated process must terminate in finite rounds.

(3) When $\epsilon = 0$, considering that the algorithm 3 produces an infinite number of iterations, the stopping termination always stands up, i.e. $\forall k, U(x_{k+1}^{\epsilon_1}, q_k^*) - U(p_k^*, y_{k+1}^{\epsilon_2}) > 0$. Without consideration

of the effect of approximate best responses' properties on the judgment of termination condition, resembling the proof of theorem 3, based on (20) and (22) we can get

$$U(p_k^*, q_k^*) \leqslant U(p_k^*, y_{k+1}^{\epsilon_2}) \leqslant U(p_k^*, y_{k+1}) + \epsilon_2 \leqslant U(p_k^*, y) + \epsilon_2, \forall y \in Y,$$

$$U(x, q_k^*) - \epsilon_1 \leqslant \max_{x \in X} U(x, q_k^*) - \epsilon_1 \leqslant U(x_{k+1}^{\epsilon_1}, q_k^*) = U(p_k^*, q_k^*), k \leqslant K_0, \forall x \in X. \tag{34}$$

hence taking limits on both sides

$$U(x, q^*) - \epsilon_1 \leqslant U(p^*, q^*) \leqslant U(p^*, y) + \epsilon_2, \forall x \in X, y \in Y, \tag{35}$$

according to (23), **every weakly convergent subsequence converges to a $\max\{\epsilon_1, \epsilon_2\}$- equilibrium**.

As the example taken in (28), not all forms of approximate best response can breach the termination condition in the whole iteration process.

Because of the fact that the strategy space $X$ is finite, combined with (21) we can get that

$$U(p_k^*, q_k^*) - U(p_k^*, y_{k+1}^{\epsilon_2}) > 0, k > K_0.$$
$$\Longrightarrow U(p_k^*, y_{k+1}) \leqslant U(p_k^*, y_{k+1}^{\epsilon_2}) \leqslant U(p_k^*, q_k^*). \tag{36}$$

Integrated with (21), easily we can get

$$U(p_k^*, y_{k+1}^{\epsilon_2}) \to U(p^*, \hat{y}^{\epsilon_2}) = U(p^*, q^*) = U(p^*, \hat{y}). \tag{37}$$

Define $\triangle_k^y \triangleq U(p_k^*, y_{k+1}^{\epsilon_2}) - U(p_k^*, y_{k+1})$. From (37) and (21), the following result can be derived:

$$\triangle_k^y \to 0, k \to \infty. \tag{38}$$

According to (36),

$$U(p_k^*, q_k^*) \leqslant U(p_k^*, y_{k+1}^{\epsilon_2}) = U(p_k^*, y_{k+1}) + \triangle_k^y \leqslant U(p_k^*, y) + \triangle_k^y, \tag{39}$$

Take limits on both sides,

$$U(p^*, q^*) \leqslant U(p^*, y). \tag{40}$$

Combined with (23), we can get that in infinite iterations, **every weakly convergent subsequence will converge to $\epsilon_1$- equilibrium**. □

**Theorem 4** *If the combinatorial optimization player employs the state-of-the-art approximate algorithm, whose computational complexity is polynomial with respect to the scale of the problem noted as $f(n, m, logK)$, the continuous adversarial player adopts LinUCB with $d$-dimension input, then the computational complexity of the algorithm is $O(p(f(n, m, logK) + Td^3))$ if $\epsilon > 0$.*

*Proof.* Providing that there exists a representative network that can compress the CO problem into $d$-dimension vector with full information, then the overall computational complexity is

$$\mathcal{O}(f(n, m, logK)) \cdot p + \mathcal{O}(Td^3) \cdot p = \mathcal{O}(p(f(n, m, logK) + Td^3)),$$

in which $T$ is the number of iterations in LinUCB, $K$ is the largest value of the single item in the CO problem. □

Note that among common algorithms for solving combinatorial optimization problems—namely, approximate algorithms, heuristic algorithms, and reinforcement learning (RL) methods—only approximate algorithms have a complexity analysis and performance guarantee. For heuristics and RL methods, complexity analysis remains an open challenge. Hence, we choose approximate algorithms as the approximate best response. Additionally, considering the continuous strategy space of the continuous player, LinUCB is an appropriate algorithm for computing its best response.

## C  PSEUDOCODE OF ALGORITHMS

The pseudocode for CCDO and CCDOA is presented in Algorithms 2 and 3, respectively.

---

**Algorithm 2** Combinatorial-Continuous Double Oracle Algorithm

---

**Input:** Game $\mathcal{G} = (X, Y, u), \epsilon \geqslant 0$.
**Output:** $(p_k^*, q_k^*)$.
 1: Initialize strategy set $X_1, Y_1$.
 2: **repeat**
 3:     Solve the mixed equilibrium $(p_k^*, q_k^*)$ in the subgame $(X_k, Y_k)$.
 4:     Find the best response $x_{k+1}, y_{k+1}$: $x_{k+1} \in \mathbb{BR}_1(q_k^*), y_{k+1} \in \mathbb{BR}_2(p_k^*)$.
 5:     $X_{k+1} = X_k \cup \{x_{k+1}\}, Y_{k+1} = Y_k \cup \{y_{k+1}\}$.
 6: **until** $U(x_{k+1}, q_k^*) - U(p_k^*, y_{k+1}) \leqslant \epsilon$

---

**Algorithm 3** Combinatorial-Continuous Double Oracle Approximate Algorithm

---

**Input:** Game $\mathcal{G} = (X, Y, u), \epsilon \geqslant 0$.
**Output:** $\sigma_k^*$.
 1: Initialize strategy set $\Pi_{1,0}, \Pi_{2,0}$.
 2: **repeat**
 3:     Solve the mixed equilibrium $(p_k^*, q_k^*)$ in the subgame $(X_k, Y_k)$.
 4:     Find the $\epsilon_1$- best response, $\epsilon_2$- best response $x_{k+1}^{\epsilon_1}, y_{k+1}^{\epsilon_2}$:
         $x_{k+1} \in \mathbb{BR}_1^{\epsilon_1}(q_k^*), y_{k+1} \in \mathbb{BR}_2^{\epsilon_2}(p_k^*)$.
 5:     $X_{k+1} = X_k \cup \{x_{k+1}\}, Y_{k+1} = Y_k \cup \{y_{k+1}\}$.
 6:     **if** $U(x_{k+1}^{\epsilon_1}, q_k^*) \leqslant U(p_k^*, q_k^*)$ **then**
 7:         $x_{k+1}^{\epsilon_1} = x$ random in $p_k^*, X_{k+1} = X_k$.
 8:     **else**
 9:         $X_{k+1} = X_k \cup \{x_{k+1}^{\epsilon_1}\}$.
10:     **end if**
11:     **if** $U(p_k^*, y_{k+1}^{\epsilon_2}) \geqslant U(p_k^*, q_k^*)$ **then**
12:         $y_{k+1}^{\epsilon_2} = y$ random in $q_k^*, Y_{k+1} = Y_k$.
13:     **else**
14:         $Y_{k+1} = Y_k \cup \{y_{k+1}^{\epsilon_2}\}$.
15:     **end if**
16: **until** $U(x_{k+1}^{\epsilon_1}, q_k^*) - U(p_k^*, y_{k+1}^{\epsilon_2}) \leqslant \epsilon$

---

## D  EXPERIMENTS PARAMETERS AND SETTINGS

### D.1  PARAMETERS OF CCDO-RL

Table 3: Parameters of CCDO-RL

| parameter | ACVRP20 | ACVRP50 | ACSP20 | ACSP50 | PG20 | PG50 |
|---|---|---|---|---|---|---|
| Iteration | 26 | 35 | 24 | 35 | 13 | 12 |
| Batchsize (prog/adv) | 512 | 512 | 512 | 512 | 1024 | 1024 |
| Prog training epoch | 10 | 25 | 10 | 20 | 150 | 150 |
| Prog training decoder | sampling | sampling | sampling | sampling | sampling | Top_p-sampling |
| Prog eval/test decoder | greedy | greedy | greedy | greedy | greedy | beam-search |
| Learning rate (prog) | 1e-4 | 1e-4 | 1e-4 | 1e-4 | 2e-4 | 2.5e-4 |
| Adv BR training epoch | 5 | 20 | 20 | 20 | 20 | 50 |
| Learning rate (adv) | 1e-4 | 5e-5 | 5e-5 | 5e-5 | 5e-5 | 5e-5 |
| Clip range (PPO in adv) | 0.2 | 0.2 | 0.2 | 0.2 | 0.2 | 0.2 |
| Value Func $\lambda$ (PPO) | 0.5 | 0.5 | 0.5 | 0.5 | 0.5 | 0.5 |
| Entropy $\lambda$ (PPO) | 0.01 | 0.0 | 0.0 | 0.0 | 0.0 | 0.0 |
| Max gradient (PPO) | 0.5 | 0.5 | 0.5 | 0.5 | 0.5 | 0.5 |

### D.2  PROBLEM SETTING

#### D.2.1  ACSP

The adversarial covering salesman problem (ACSP) is one variant of the traveling salesman problem (TSP) with adversaries. The biggest difference is that each city $i$ has one coverage radius $r_i$. If some unvisited cities are covered by visited cities, then these unvisited are seen as being visited in the TSP. Hence the salesman in the ACSP aims to find the shortest path such that all cities have been visited or covered. However, due to some external factors like transportation situations, the coverage radius may be influenced. Similar to the influence model in (7), the real coverage radius is

$$r_i = \hat{r}_i + \sum_{m,n=1}^{K} \phi_{i,m}\phi_{i,n}y_{mn}, \tag{41}$$

where $\phi_{i,m}$ is the $m_{th}$ element of $\phi_i$, i.e. the transportation vector at the city $i$ correlated with the city's location, i.e. $\phi_i = \frac{1}{2}\phi \cdot loc_i$ where $\phi$ is one constant $K \times 2$ matrix. and $y_{mn}$ is the component of the environmental parameter matrix $y$ controlled by the adversary. Hence the objective function is like the ACVRP,

$$\min_{x \in X_{csp}} \max_{y \in [0,1]^9} length(x), \tag{42}$$

where $length$ is the summary distance from the start to end (two cities can be different), and the path $x$ should visit or cover all cities.

#### D.2.2  ACVRP

The adversarial capacitated vehicle routing problem (ACVRP) is that there is one depot and one vehicle with constrained good capacity which starts and ends at the depot where the vehicle can supplement goods. The objective of this vehicle is to find the shortest path while satisfying all the demands of customers on the map. Each customer $i$ owns its two-dimension position $(x, y)$ and an estimated demand $\hat{d}_i$. According to the influence model (7), the real demand $d_i$ is set as follows:

$$d_i = \hat{d}_i + \sum_{m,n=1}^{K} \omega_{i,m}\omega_{i,n}y_{mn}, \tag{43}$$

in which $\omega_{i,m}$ is the $m_{th}$ element of $\omega_i$, i.e. the weather vector at the customer point $i$, and $\alpha_{mn}$ is the component of the environmental parameter matrix $\alpha$ controlled by the adversary. In our setting,

we assume that the weather condition is related to the customer's location, i.e. $\omega_i = \frac{1}{2}\omega \cdot loc_i$ where $\omega$ is one constant $K \times 2$ matrix. Before the vehicle chooses the customer to deliver goods, it can only know the estimated demand $\hat{d}_i$. When arriving at the chosen customer, the vehicle knows the real demand of this customer. It should be noted that goods can't be split up. In other words, if the remaining capacity of goods can't satisfy the real demand of the chosen customer, the vehicle should come there again until its current capacity meets the real demand $d_i$.

Hence the objective function of the vehicle is to minimize the maximal path $x$ under the changeable environment parameters $y \in [0, 1]^9$, meeting all the demands of customers

$$\min_{x \in X_{cvrp}} \max_{y \in [0,1]^9} length(x), \tag{44}$$

where $length$ is the summary distance from the start to end, adopting the Euclidean distance.

### D.2.3 PG

About the PG, we set it as the classical security patrolling game with two players: one defender and one attacker. There are $N$ targets to protect, each one has an individual prize $v_i$ and an estimated attack probability $\hat{p}_i$ defined by objective factors. The defender tries to find a path to maximize the cumulative prize attained from some targets prevented from attacks successfully under the total distance constraint. For the attacker, it will decide the real attack probability of each target based on the estimated probability $\hat{p}_i$ and its objective is to reduce the cost of being caught, equivalent to reducing the total patrolling revenue of the defender. Identically, the attack probability on each target also follows the influence model in (7),

$$p_i = \hat{p}_i + \sum_{m,n=1}^{K} \xi_{i,m}\xi_{i,n}y_{mn}, \tag{45}$$

where $\xi_i$ is the attacker's preference vector to the target $i$, to keep consistent with the two settings before, we still set the preference vector as related to the location, i.e. $\xi_i = \frac{1}{2}\xi \cdot loc_i$ where $\xi$ is one constant $K \times 2$ matrix. The objective function of this security game is

$$\max_{x \in X_{op}} \min_{y \in [0,1]^9} \sum_{i=1}^{N} p_i v_i \mathbb{I}_{\Pi}, \tag{46}$$

where $\Pi$ is the set of patrolled targets on the path $x$.

### D.3 ATTENTION MODEL & HYPERPARAMETERS

#### INSTANCE GENERATION

For ACVRP and ACSP, we use the default data generated from RL4CO (Berto et al., 2023). For PG, we use the instances generated from a generator provided in the AI for TSP competition hosted at IJCAI21 [1].

#### MODEL ARCHITECTURE

For the protagonist of three COPs, we use the REINFORCE algorithm with the Attention network used by possessing the graph information in RL4CO. About decoders, we use the static embedding for ACVRP and PG provided in RL4CO (Berto et al., 2023), and a dynamic embedding for ACSP from (Li et al., 2021).

About the encoder, three instances all adopt the attention network with 8 heads while the number of the network layer is 3 for ACSP and ACVRP, and 5 for PG. Different COPs have their own state and context of the environment, input details for three COPs are as follows:

- **ACVRP:** The environment context is each customer's location, demand, and weather vector. The state at time slot $t$ is the current location and remaining capacity of the vehicle.

---

[1] https://github.com/paulorocosta/ai-for-tsp-competition

- **ACSP:** The environment context is each city's location, estimated coverage, and transportation vector. The state at time slot $t$ is the current location( the chosen node to visit at the last time slot) and the start point.
- **PG:** The environment context is each target's location, estimated attack probability, and prize. The state at time slot $t$ is the start point and the current location of the defender.

# E DISCUSSION ON SCALABILITY AND POTENTIAL OPTIMIZATION OF CCDO-RL

## E.1 SCALABILITY ANALYSIS OF CCDO-RL

In CCDO-RL, three components need to be trained or computed:

1. The combinatorial player's policy. This player solves a combinatorial optimization problem (COPs) under a specific strategy of the adversary.
2. The continuous player (as the adversary) with an infinite continuous strategy space.
3. The computation of Mixed Nash Equilibria (NE) in the subgame.

Next, we will analyze the computation time for each component individually, from both theoretical and experimental perspectives. For the experimental part, we will use the 50-node Patrolling Game (PG) scenario, which is the most challenging problem instance in our experiments, as an example.

1. The combinatorial player is trained using Graph Neural Networks (GNN) and REIN-FORCE to find feasible and optimal solutions for NP-complete COPs. This complexity requires reinforcement learning to invest more time and data for effective model training. In the experiment, training a stable and high-performing combinatorial model takes 26 minutes (10000 data, 1024 batch size, 150 epochs) with the continuous player fixed.
2. The continuous player is trained by PPO to tackle a one-step problem with a continuous objective function building on strategies of the combinatorial player. It still utilizes GNN to understand graph structure. One action per episode reduces training times compared to the combinatorial player while achieving similar approximate error levels. In our experiments, training a high-performing model takes only 4 to 5 minutes (10000 data, 1024 batch size, and 50 epochs), less than one-fifth of the time required for the combinatorial player.
3. For the NE solution, the mixed equilibria in a zero-sum game can be solved by the linear programming method which has polynomial complexity in the size of the game tree. From the perspective of theoretical complexity and experiment implementation, the computational time is negligible (less than 2s).

From the statement above, we can conclude that more than five-sixths of the computation time is spent training the model or strategy of the combinatorial player. Therefore, a crucial aspect of addressing the scalability issue is to enhance the speed of solving the Combinatorial Optimization Problems (COPs) using Reinforcement Learning (RL).

## E.2 POTENTIAL OPTIMIZATION OF SCALABILITY

In this subsection, we briefly discuss two main aspects as potential ways of improvement.

**COPs Simplification Method**

1. The pruning method: this one was introduced in the original scale of COPs to reduce the number of possibly useful actions. In this way, the computational burden will be decreased (Manchanda et al., 2019; Lauri et al., 2023).
2. Broken down into subproblems: in some concrete COPs like TSP (Fu et al., 2021), and VRP (Hou et al., 2023), the originally large-scale problem can be broken down into smaller problems to solve, thereby reducing the solution difficulty.

**RL algorithms**

1. Learning Time Reduction: increase the sampling data quality by attaining good-performance data from pre-trained RL models or heuristic algorithms on COPs (seemingly like the model-based RL).

2. NN Model Adjustment: most constructive neural network fitting combinatorial optimization can not solve problems with large-scale instance sizes. One feasible way is to design an NN model with strong scalability which means that the trained model on small-scale problem instances can be used on large-scale ones, such as in influence maximization (Chen et al., 2023).

3. Distributed training: reduces the time required for training by splitting the computational workload across multiple devices.

### E.3 EXPERIMENT RESULTS OF CCDO-RL'S SCALABILITY

We test the CCDO-RL model (trained on 50-node graphs) on larger CSP and PG scenarios. On unseen 100-node and 200-node graphs (100 of each type), CCDO-RL outperformed other baselines while requiring significantly less test time compared to the heuristic algorithm (especially in CSP), as demonstrated in Tables 4.

Table 4: Scalability results on ACSP and PG (smaller is better in ACSP, larger is better in PG)

| method | CSP | | PG | |
|---|---|---|---|---|
| | 100 nodes | 200 nodes | 100 nodes | 200 nodes |
| Heuristic | 7.38 (5h 46mins) | 6.95 (7h 16mins) | 7.71 (53s) | 11.01 (120s) |
| RL against Stoc | 7.34 | 9.86 | 7.83 | 9.24 |
| CCDO-RL | **4.61** | **4.89** | **8.42** | **11.07** |

## F ADDITIONAL EXPERIMENTAL RESULTS

### F.1 CONVERGENCE TO NE IN 50-NODE GRAPHS

All experiments on three COPs are implemented in Python and conducted on two machines. One is NVIDIA GeForce RTX 4090 GPU and Intel 24GB 24-Core i9-13900K. The other is NVIDIA V100 GPU and Inter 256 GB 32-Core Xeon E5-2683 v4.

Illustrated by Fig. 2, we observe that CCDO-RL also converges close to the real NE in 35 iterations for ACSP and ACVRP, and for PG it takes 12 iterations. Their runtimes are 10h 20mins, 4h 40 mins, and 9h 6mins respectively. Exploitability of ACSP, ACVRP, and PG are 0.06, 0.27, and 0.13 respectively. The phenomenon that exploitabilities on three COPs of 50 nodes are all larger than that on the 20-node map is reasonable and acceptable because the hardness of solving solutions grows exponentially on NP-hard problems, such as these three.

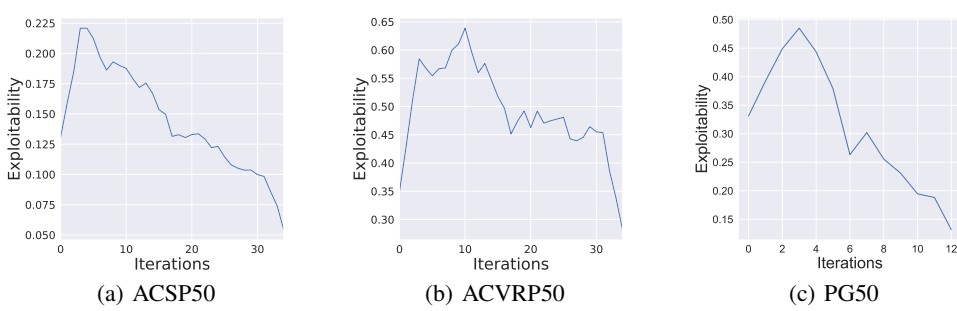

| (a) ACSP50 | (b) ACVRP50 | (c) PG50 |

Figure 2: Exploitability on Three COPs of 50 Nodes

### F.2 FULL RESULTS OF CCDORL

Here we replenish and analyze results against the stochastic adversary on three COPs. "Adv" or "no adv" columns in following tables indicates whether all instances are influenced by the learned adversary or a random adversary, respectively.

From "no adv" columns in Tables 5 and 6 (ACSP), we can see the average reward (seen graphs) and generalizability (unseen graphs) of the combinatorial player trained in CCDO-RL are both better than others, even though the RL baseline is trained against the stochastic adversary solely. The average improvement against RL baseline is 3.96% on different types of graphs and different nodes. Similarly under the ACVRP and PG settings, average improvements against are 3.88% and 2.72% respectively. We can find CCDO-RL can also get the better reward under the usual stochastic setting, not just the adversarial setting.

Table 5: Full results on ACSP in 20 nodes (smaller values are better)

| method | seen graphs | | unseen graphs | |
| --- | --- | --- | --- | --- |
| | no adv | adv | no adv | adv |
| Heuristic | 6.17±1.23 | 6.13±1.20 | 6.03±1.30 | 6.20±1.33 |
| RL against Stoc | 3.39±0.46 | 3.50±0.47 | 3.39±0.46 | 3.56±0.37 |
| CCDO-RL | **3.18**±0.44 | **3.25**±0.42 | **3.19**±0.41 | **3.31**±0.35 |

Table 6: Full results on ACSP in 50 nodes (smaller values are better)

| method | seen graphs | | unseen graphs | |
|---|---|---|---|---|
| | no adv | adv | no adv | adv |
| Heuristic | 7.50±1.50 | 7.55±1.42 | 7.57±1.49 | 7.60±1.37 |
| RL against Stoc | 4.29±0.61 | 4.55±0.62 | 4.20±0.56 | 4.57±0.58 |
| CCDO-RL | **4.16**±0.48 | **4.31**±0.51 | **4.17**±0.48 | **4.39**±0.52 |

Table 7: Full results on ACVRP in 20 nodes (smaller values are better)

| method | seen graphs | | unseen graphs | |
|---|---|---|---|---|
| | no adv | adv | no adv | adv |
| Heuristic | 7.50±1.36 | 7.65±1.23 | 7.74±1.30 | 7.64±1.30 |
| RL against Stoc | 7.68±1.32 | 7.55±1.16 | 7.70±1.30 | 7.67±1.30 |
| CCDO-RL | **7.43**±1.26 | **7.42**±1.21 | **7.62**±1.30 | **7.55**±1.28 |

Table 8: Full results on ACVRP in 50 nodes (smaller values are better)

| method | seen graphs | | unseen graphs | |
|---|---|---|---|---|
| | no adv | adv | no adv | adv |
| Heuristic | 13.22±1.75 | 13.38±1.70 | 13.45±1.67 | 13.27±1.87 |
| RL against Stoc | 13.89±1.85 | 13.90±1.63 | 13.95±1.70 | 13.85±1.53 |
| CCDO-RL | **13.14**±1.72 | **13.28**±1.52 | **13.14**±1.72 | **13.15**±1.59 |

Table 9: Full results on PG in 20 nodes (larger values are better)

| method | seen graphs | | unseen graphs | |
|---|---|---|---|---|
| | no adv | adv | no adv | adv |
| Heuristic | 2.70±1.15 | 2.64±1.03 | 2.64±1.18 | 2.43±0.98 |
| RL against Stoc | **2.81**±1.25 | 2.71±0.90 | 2.71±1.35 | 2.50±0.95 |
| CCDO-RL | 2.75±1.06 | **2.75**±0.87 | **2.77**±1.19 | **2.56**±0.92 |

Table 10: Full results on PG in 50 nodes (larger values are better)

| method | seen graphs | | unseen graphs | |
|---|---|---|---|---|
| | no adv | adv | no adv | adv |
| Heuristic | 4.69±1.81 | 4.53±1.84 | 4.47±2.02 | 4.19±1.69 |
| RL against Stoc | 4.87±2.75 | 4.80±2.18 | 4.58±2.42 | 4.26±2.17 |
| CCDO-RL | **5.12**±1.97 | **5.01**±1.91 | **4.84**±2.16 | **4.70**±1.94 |

