# OpenReview forum: "Can Reinforcement Learning Solve Asymmetric Combinatorial-Continuous Zero-Sum Games?"
_ICLR.cc/2025/Conference — ICLR 2025 Poster_

### Official Review · Reviewer_LGc6 · 2024-10-26

**Soundness:** 2
**Presentation:** 3
**Contribution:** 3
**Rating:** 6
**Confidence:** 3

**Summary:**

This paper introduces and studies a new class of zero-sum games – two-player Asymmetric Combinatorial-Continuous zEro-Sum (ACCES) games, where one player has a combinatorial action space whereas the other has an infinite compact space. The asymmetry lies in the differing nature of the players’ strategy spaces. The authors claim that ACCES games closely resemble real-world problems, particularly in combinatorial optimization problems (COPs), min-max games, and security games. To evaluate their algorithms, the authors provide three different ACCES game scenarios: adversarial covering salesman problem (ACSP), adversarial capacitated vehicle routing problem (ACVRP), and patrolling game (PG).

 As with any new problem in game theory, the authors first prove the existence of the Nash Equilibrium (NE) for this game. The authors then propose a double oracle-based algorithm called Combinatorial Continuous DO (CCDO) to solve ACCES games, alongside proving the convergence of the algorithm. Finally they propose a Reinforcement Learning algorithm, CCDO-RL with convergence guarantees and empirically demonstrate the effectiveness of the proposed algorithm. CCDO-RL adopts RL to compute the approximate best responses.

**Strengths:**

- The paper tackles a novel problem in 2p0s games by considering asymmetry in strategy space rather than the common forms of asymmetry (e.g., information asymmetry).
- The authors provide theoretical proofs for the existence of NE and convergence of their algorithms, as well as empirical demonstrations of their effectiveness and superiority compared to baselines based on heuristics and a single-agent RL algorithm.

**Weaknesses:**

- The patrolling game in the introduction might benefit from a better explanation, especially labels for P1 and P2 (attacker/defender). It would be helpful if the authors could label which player is the attacker and which is the defender.
- Algorithm 1 in the paper seems very similar to the XDO/NXDO algorithms (McAleer et al., 2021). Perhaps the novelty is in computing the BRs (McAleer et al. consider BRs but the authors here consider approx. BRs), but the underlying algorithm, as presented, seems the same.  Highlighting the difference of the proposed algorithm with existing algorithm such as XDO/NXDO might underscore the challenge of solving the proposed problem.
- How the mixed equilibria (step 3 in all algorithms) are solved in the subgame is unclear. This is likely the crucial part of the problem, given the different strategy spaces associated with the players. Briefly explaining the computational technique would be helpful.

**Questions:**

* The dynamics of the ACCES game is unclear. Is it a turn based or a simultaneous move game? Also, is it a fixed-horizon game or an infinite one with some termination criterion?
* In line 166, shouldn't $X$ be **all** *routes* instead of **any** ?
* Line 140:
> As far as we know, they are limited to matrix games, ...

  * Don't McAleer et al. (2021) consider both extensive-form games and a continuous game? I am not sure what is being referred to as being "limited to matrix games" as McAleer et al. (2021) also prove convergence to $\varepsilon$-NE. Could authors please elaborate on this? Perhaps I misunderstood.

* Line 328:
> However, the approximation of BR may cause circumstances where the utility of the approximate best response is lower than that of NE in the subgame"

  * I am curious as to how this would impact the final policy. Yes, it is possible that the approximate BR may not be accurate leading to a negative NashConv, which is theoretically not possible. Could authors comment on the impact of removing the steps 6-9 in Algorithm 1 from purely experimental standpoint? Also, explaining this part in a bit detail might help readers understand the issue.

* I couldn't find the results and/or discussion on "how different ABRs influence convergence". Perhaps my interpretation of different ABRs as different BR approximating algorithms is not correct. Could authors please comment on this?

*McAleer, S., Lanier, J. B., Wang, K. A., Baldi, P., & Fox, R. (2021). XDO: A double oracle algorithm for extensive-form games. Advances in Neural Information Processing Systems, 34, 23128-23139.*

---

> ### Author Response · Authors · 2024-11-21
> **Weaknesses**
>
> We thank the reviewer for the very detailed and constructive comments and feedback!
>
> **W1: Clarified explanation for the patrolling game**
>
> Thanks for your advice. We have updated this example as follows (Section 1, Paragraph 4):
>
> Player 1’s strategy space is combinatorial, while Player 2’s is infinite and compact with a continuous utility function. As an illustrative example (more examples in Section \ref{sec_exp}), we consider a patrolling game between a defender (Player 1) and an attacker (Player 2). To prevent attacks from the attacker, the defender chooses a feasible route to patrol a subset $\Pi$ of all targets $\{1, 2, …, N\}$ meanwhile satisfying the total distance constraint $L_{all}$ because of limited patrol time. For the attacker, the strategy is the attack probability vector $\{p_1, p_2, …, p_N\}$ for the target set. Besides, each target $i \in \{1, 2, …, N\}$ has its own value $v_i$. The utility function for the defender is the expectation of successfully protected target values, i.e. $ U_d = \sum_{i=1}^N p_i v_i \mathbb{I}_{\Pi}$. The attacker’s utility function is then: $U_a = - U_d$.
>
> **W2: Differences of CCDO-RL to XDO/NXDO**
>
> Thanks for your positive advice! In Related Work and Section 5.1, we have added and highlighted the difference between our algorithm and DO’s variants.
>
> Compared with XDO/NXDO, our algorithm is different from the perspective of convergence analysis of CCDO and CCDO-RL, and termination criterion. The detailed difference is as follows:
>
> 1. Novelties of convergence analysis of CCDO and CCDO-RL
>
>   * XDO is a tabular extensive-form double oracle algorithm, while NXDO extends it using deep reinforcement learning (DRL). It guarantees convergence **only in matrix games**, as its strategy space can be expanded to the full game within a finite number of iterations which does not work for the **infinite/continuous strategy space** in ACCES games. NXDO addresses continuous-action games empirically with Deep RL, though it lacks algorithmic guarantees. Our algorithm, CCDO-RL (CCDOA), has a convergence analysis (Theorem 3) in ACCES games, which is not feasible for XDO or other DO variants like ODO, despite a similar framework. In CCDO-RL, the DRL component helps achieve approximate best responses, leveraging its strengths in solving combinatorial problems (Section 5.2, Lines 336-342), rather than facing issues with continuous strategy spaces as in XDO.
>   * We propose the convergence analysis with approximate best responses (ABRs) and **different ABRs’ influence on the convergence**. Approximate best responses (ABR) are very commonly used in COPs due to their NP-hardness. It’s therefore critical to consider its effect on the convergence of ACCES games which wasn’t addressed before. We provide the novel convergence analysis of CCDOA\CCDO-RL and study different ABR’s influence on convergence (Theorem 3 Item 2 and Remark 2) in Section 5.4.
>
> 2. Termination criterion
>
> It's interesting to note that XDO doesn't have a specific termination criterion because it can naturally stop because of **the finiteness of actions but cannot be possibly guaranteed in ACCES games** because of the **infinite strategy space** of the continuous player (Player 2). Due to the continuity/infiniteness of Player 2’ strategy space, we've incorporated a termination criterion (Line 11 of Algorithm 1). This helps us ensure that the algorithm can stop while still maintaining convergence.
>
> **W3: Solving mixed NE in the subgame**
>
> Thanks for your constructive suggestion! We have added the statement of solving mixed NE in Section 5.2, paragraph 2.
>
> We solve for the mixed Nash equilibrium in the ACCES game using the support enumeration algorithm [1]. This approach is based on linear programming and utilizes the Nashpy implementation [2], which relies solely on the utility matrix of the subgame without considering the strategy spaces.
>
> [1] Tim Roughgarden. Algorithmic game theory. Communications of the ACM, 53(7):78–86, 2010.
>
> [2] Vincent Knight and James Campbell. Nashpy: A python library for the computation of Nash equilibria. Journal of Open Source Software, 3(30):904, 2018.

---

> > ### Author Response · Authors · 2024-11-21
> > **Questions**
> >
> > **Q1: Dynamics of the ACCES game**
> >
> > Thanks for your question. The dynamic of the ACCES game is a simultaneous move and infinite game with a termination criterion (shown in the last line of all algorithms). We have supplemented this in Section 1, paragraph 4.
> >
> > **Q2: Grammar Error**
> >
> > Thanks for pointing out. We have corrected this in the updated version.
> >
> > **Q3: Difference with XDO**
> >
> > Thanks for your comment. We have clarified this sentence in Related Work as follows,
> >
> > ‘They are all limited to matrix games in theories related to the existence and convergence of NE although \citet{mcaleer2021xdo} conducts experiments on continuous-action games by Deep RL.’
> >
> > From W1, we know that McAleer et al. (2021) don’t provide the convergence guarantee of the continuous-action game, they only experimented with the Loss Game by DRL.
> >
> > **Q4:  Lines 6-9 in Algorithm 1**
> >
> > Good question! Thank you for bringing this up. We have added the explanation of Lines 6-9’s experimental influence in Section 5.2, paragraph 1.
> >
> > From an experimental standpoint, adding a strategy that leads to a negative NashConv to the subgame can **increase both computational and memory burden**. In each round, we need to save ABR’s strategy/model and compute its utility value against the other player’s every strategy in the subgame to complete the utility matrix (used in the computation of mixed NE). So introducing the unnecessary policy to the subgame will prolong the computation time (complementation of utility matrix and computation of mixed NE) in each round. Besides, this may even extend the number of iteration rounds, as it affects the computation of mixed Nash Equilibria.
> >
> > **Q5: Discussion on ABR**
> >
> > Thanks for your comment. We have rewritten and highlighted this point in Section 5.3. Our explanation is as follows.
> >
> > We mainly focus on the influence of ABRs with **different degrees of approximation** on the number of algorithm inner iterations. The infinite strategy space for Player 2 can potentially lead to an infinite number of iterations when $\epsilon = 0$ (as discussed in Theorem 2, Item 1, and Theorem 3, Item 3). However, **if the ABR meets certain conditions, the algorithm will converge in a finite number of iterations, even when the stopping criterion $\epsilon = 0$**. In our work, we propose two new conditions. The first is that the ABR for Player 2, which has an infinite continuous strategy space, has a uniform lower bound (see Theorem 3, Item 2, and further explanation in Lines 385-387). The second condition relates to the absolute approximate degree of the ABR, whether for Player 1 or Player 2. We believe these conditions provide deeper insights and offer convergence guarantees that may be useful for readers who wish to apply other algorithms to compute the ABR.

---

> > > ### Comment · Reviewer_LGc6 · 2024-11-21
> > >
> > > Thank you for addressing my concerns.
> > >
> > > I have a few followup questions:
> > >
> > > 1. Since the subgame is a matrix game, doesn't that imply that the action spaces are discrete (with each row being, say, P1's action and column being P2's action)? I am a little confused, perhaps you could shed some light?
> > >
> > > 2. Regarding the termination criterion, I was actually referring to the termination criterion of the game and not the algorithm. Say both players play arbitrary random action throughout the course of the game, when would the game end? For example, in pursuit-evasion game, the game would end when the pursuer captures the evader.
> > >
> > > Other than these two, I think most of my concerns are addressed.

---

> > > > ### Author Response · Authors · 2024-11-22
> > > > **Follow-up Questions**
> > > >
> > > > Thanks for your prompt responses.
> > > >
> > > > 1. The key idea of CCDO/CCDO-RL is that the mixed NE of the subgame can converge to the mixed NE of the original game via repeated iterations. The subgame being a matrix game does not imply that the original game is also a matrix game. On a high level, the subgame is **only an approximation of the original game**, which can be sort of viewed as a “discretized” version of the original game. The subgame is incrementally updated by adding the best responses so that it gets closer to the original game, but it is still just an approximation.
> > > >
> > > > 2. Note that the ACCES game is **a strategic (static) game**. There is no explicit time dimension which means that all decisions are made at the same moment. Once each player chooses their strategy, then the game terminates.

---

> > > > > ### Comment · Reviewer_LGc6 · 2024-11-22
> > > > >
> > > > > Thank you for the clarification, specially about the game being static and not an extensive form type game.
> > > > >
> > > > > I understand that the idea is to incrementally add the best responses to the subgame so that after enough iterations it gets closer to the original game. I just wanted to make sure that the actions are indeed discretized in the subgame.

---

> > > > > > ### Author Response · Authors · 2024-11-22
> > > > > >
> > > > > > Thanks for confirming with us. We really appreciate your quick replies. We have clarified the dynamic of the ACCES game in the updated version. Please feel free to let us know if you have any further questions!

---

> > > > > > > ### Comment · Reviewer_LGc6 · 2024-11-22
> > > > > > >
> > > > > > > Thank you for addressing my questions and concerns.
> > > > > > >
> > > > > > > I will raise my score to a 6.

---

> > > > > > > > ### Author Response · Authors · 2024-11-22
> > > > > > > >
> > > > > > > > We sincerely appreciate your constructive feedback and the improved score. Your comments were instrumental in helping us enhance our work.

---

### Official Review · Reviewer_xkfi · 2024-10-28

**Soundness:** 4
**Presentation:** 4
**Contribution:** 3
**Rating:** 8
**Confidence:** 4

**Summary:**

In this paper, the authors define and study a new useful class of two-player games called ACCES games, which feature asymmetry between the players' strategy spaces. One player has a combinatorial strategy space, while the other has a continuous one. The paper has 3 main contributions. First, the authors prove the existence of a Nash equilibrium for two-player ACCES games. Second, they describe an extension to the Double Oracle method that solves two-player ACCES games by provably converging to Nash. This method (CCDO) relies on exact best response computation at every iteration.  Finally, they present a modified version of CCDO that uses RL to approximate best responses and thus approximate Nash in more practical ACCES settings.

**Strengths:**

* I like this formalization. It is intuitive and seems to be a natural extension of the two-player zero-sum symmetric case that fits many interesting scenarios (like the nature vs. player and patrolling games examples mentioned in the paper).
* The paper is impressively well-written and easy to follow. The main contributions are clearly stated and motivated early on, and the theoretical results are presented in a digestible manner that guides the reader through them.
* The authors present theoretically sound algorithms for solving and approximately solving these games. These results are impressive on their own, but I believe they are valuable to the community because they could act as a foundation for developing more scalable algorithms in this domain in the future.

**Weaknesses:**

* In the conclusion, the authors mention that scalability may be a limitation of their work. Though it's not the main point of the paper, I'm somewhat concerned that the scalability of the approximate best response may limit the applicability of CCDO-RL. I don't believe the authors have to necessarily evaluate their methods on larger domains, but some insight on how the RL component of every iteration scales could help alleviate these concerns.
* The convergence guarantees of CCDO-RL seem dependent on finding $\epsilon$ best-responses at every iteration. To the previous point, this may be unrealistic in some larger domains. This somewhat weakens the guarantees of the algorithm, but I understand that this seems true for many algorithms whose dynamics depends on best response approximation.

**Questions:**

* What specific scalability concerns do the authors have with CCDO-RL?
* Is CCDO-RL guaranteed to terminate? It seems if both conditions on lines 6 and 8 in Algorithm 1 fail, then the strategy set is unchanged.

**Additional Comments**
* There seems to be a mistake on line 71 regarding the attacker and defender's utilities in the example patrolling game.
* From the formalization, it is clear that every two-player ACCES game is zero-sum, but I suggest mentioning that earlier.

---

> ### Author Response · Authors · 2024-11-21
>
> We thank the reviewer for the positive and constructive comments.
>
> **W1: Scalability**
>
> Great point. As noticed by the reviewer, scalability is not the main focus of the paper, but we do agree that this is a critical aspect that needs further work. To shed some light on this point, we have added the following discussion to Appendix E.2.
>
> [COPs simplification method]
> * **The pruning method**: this one was introduced in the original scale of COPs to reduce the number of possibly useful actions. In this way, the computational burden will be decreased [1, 2].
> * **Broken down into subproblems**: in some concrete COPs like TSP[3], and VRP [4], the originally large-scale problem can be broken down into smaller problems to solve, thereby reducing the solution difficulty.
>
> [RL algorithms]
> * **Learning Time Reduction**: increase the sampling data quality by attaining good-performance data from pre-trained RL models or heuristic algorithms on COPs (seemingly like the model-based RL).
> * **NN Model Adjustment**: most constructive neural network fitting combinatorial optimization can not solve problems with large-scale instance sizes. One feasible way is to design an NN model with strong scalability which means that the trained model on small-scale problem instances can be used on large-scale ones, such as in influence maximization [5].
> * **Distributed training**: reduces the time required for training by splitting the computational workload across multiple devices.
>
> **W2: Convergence guarantee dependent on finding ϵ best-responses**
>
> Thanks for your comment. We really agree with the reviewer. Find best responses(BRs) or $\epsilon$- best responses are an important part of all DO-based algorithms because we need to assume the property of approximate BRs to prove the convergence guarantee and the approximate degree to NE. To some extent, \epsilon-best response plays a very important role in the whole algorithm.
>
> **Q1: Scalability**
>
> Please see the response to W1.
>
> **Q2: Termination guarantee of CCDO-RL**
>
> Good question! CCDO-RL had its convergence guarantee as stated in Theorem 3, which considers RL as a method for achieving an approximate best response(ABR).
>
> Regarding the scenario where both conditions on Lines 6 and 8 fail, we think it’s a special case of Theorem 3 Item 1, i.e. CCDO-RL converges to the $(\epsilon_1 + \epsilon_2)$- equilibrium. Two ABRs have their approximate error bound, $\epsilon_1$ and $\epsilon_2$ respectively.  If Lines 6 and 8 fail together, assuming the current subgame mixed NE is $(p_k^*, q_k^*)$, we can get that
>
> $$\max_{x \in X}U(x, q_k) - \min_{y \in Y}U(p_k, y) \leq \epsilon_1 + \epsilon_2.$$
>
> Set $\bar{\epsilon} = \epsilon_1 + \epsilon_2$ as a new stopping criterion in CCDO, we can draw our conclusion by Theorem 2 Item 2.
>
> **Additional Comments**
>
> Thanks for your comments! We have corrected these two points in the updated version.
>
> [1] Manchanda S, Mittal A, Dhawan A, et al. Learning heuristics over large graphs via deep reinforcement learning[J]. arXiv preprint arXiv:1903.03332, 2019.
>
> [2] Lauri J, Dutta S, Grassia M, et al. Learning fine-grained search space pruning and heuristics for combinatorial optimization[J]. Journal of Heuristics, 2023, 29(2): 313-347.
>
> [3] Fu Z H, Qiu K B, Zha H. Generalize a small pre-trained model to arbitrarily large tsp instances[C]// AAAI. 2021, 35(8): 7474-7482.
>
> [4] Hou Q, Yang J, Su Y, et al. Generalize learned heuristics to solve large-scale vehicle routing problems in real-time[C]// ICLR. 2023.
>
> [5] T. Chen, S. Yan, J. Guo, and W. Wu, "ToupleGDD: A Fine-Designed Solution of Influence Maximization by Deep Reinforcement Learning," in IEEE Transactions on Computational Social Systems, vol. 11, no. 2, pp. 2210-2221, April 2024.

---

> > ### Comment · Reviewer_xkfi · 2024-11-26
> >
> > Thank you for your response and for addressing my questions and concerns. I will maintain my score.

---

> > > ### Author Response · Authors · 2024-11-28
> > >
> > > Thanks for the reviewer’s valuable suggestions and positive views. Please feel free to share any further thoughts or questions you might have.

---

### Official Review · Reviewer_ptbk · 2024-11-02

**Soundness:** 3
**Presentation:** 3
**Contribution:** 2
**Rating:** 6
**Confidence:** 3

**Summary:**

The paper defines and studies a novel class of two-player zero-sum games termed ACCES (Asymmetric Combinatorial-Continuous zEro-Sum) based on the premise that most zero-sum games that have been studied are either symmetric or confined to matrix games (whenever asymmetric). A natural motivation for this class is a player with a combinatorial action space (e.g., path minimization, scheduling) who plays against an environment that adversarially sets its parameters drawing from continuous parameter-value spaces (e.g., customer demand, edge weights, targets etc). The paper seeks to answer three motivating in this setting: whether a Nash equilibrium (NE) exists, whether it can be found and whether it can be found efficiently and provides affirmative answer to all three. The existence of NE is established through the properties of weakly sequential compactness and continuity of expected utility function that are established for these games. Regarding the other two questions, the paper develops an algorithm that is based on the double-oracle (DO) methods that have been used to solve finite zero-sum games and proposes an approximate, RL-based variant to tackle the exponential blow-up in computing exact best responses in the combinatorial action space of the first player. The paper complements the theoretical proofs of convergence with experiments in three ACCES environments, also demonstrating improved convergence performance against baseline algorithms.

**Strengths:**

- The paper is well-written, clearly explains its motivation and its research questions. It also provides a fair discussion of its limitations, mainly the scalability of its proposed algorithms.
- The paper is rigorous with proofs of convergence that seem correct to the extent that I could verify.

**Weaknesses:**

- While the ACCES games are interesting and cover various settings as demonstrated by the studied environments, I would have expected some more machine-learning/neural network motivation when I read about the "adversarial parameter setting of the environment" that was ultimately missing from the paper. Some examples that may be helpful: (https://arxiv.org/abs/2003.01820, https://arxiv.org/abs/2002.06673).

- The paper does an effort to acknowledge related work, however literature on zero-sum games is vast and recent papers on learning in zero-sum (matrix) games are not entirely acknowledged (e.g., Online Learning in Periodic Zero-Sum Games and references therein, Zero-sum polymatrix games, Efficiently Computing Nash Equilibria in Adversarial Team Markov Games, The complexity of constrained min-max optimization, https://arxiv.org/abs/2011.00364 etc).

- I found the existence result not surprising (maybe I am missing some technically difficult step?) and immediately following from the existence in both continuous (and bounded) and finite zero-sum games. I have a similar consideration for the convergence of the DO-based algorithm.

- Essentially combining my two previous points, it is not entirely clear to me how novel this environment is and how much more interesting that the min-max optimisation in non-convex, non-concave settings. I think that the paper does not make a very convincing argument that the current setting is not a derivative, not-niche and sufficiently different and more complex setting than what is currently studied in the literature.

**Questions:**

Can the authors discuss/address the weaknesses mentioned above?

**Post-rebuttal**: I thank the authors for their responses. I still believe that the properties of weak sequential compactness and continuity of the expected utility function are fairly straightforward and hence that both the existence of Nash equilibria in ACCES games and the class itself don't really depart much from existing classes of zero-sum games. Also, while the updated version cites some zero-sum papers, I don't see a meaningful discussion/comparison with the min-max optimisation in non-convex, non-concave games or these existing classes of zero-sum games. I think this discussion requires more thorough study of the related works which cannot be done on the fly during the limited time of this discussion - and is also beyond my capacity to engage in such a discussion. This limits the contribution of the current paper in my opinion.

Nevertheless, I appreciate the practicality of the algorithms for real-world cases that the paper offers and which I had underestimated in my original review. I have revised my scores accordingly.

---

> ### Author Response · Authors · 2024-11-21
>
> We sincerely appreciate your constructive feedback on our work.
>
> **W1: ML (RL)/neural network motivation**
>
> Thanks for your comments. We have added the adversarial motivation in Section 5.2. We will explain the motivation of the adversary from two perspectives, solvability to diverse instances of the problem, and generalizability.
>
> * **Solvability to diverse instances of the problem**: RL combined with GNN demonstrates strong adaptability in handling diverse instances of a problem. For example, in scenarios like the patrolling game, where target positions and values vary, RL+GNN effectively updates its strategy and makes good decisions. This is because it learns **the complex nonlinear mapping** from problem-specific information—often represented as high-dimensional graph data in combinatorial optimization problems (COPs)—to precise decision-making actions.
> * **Generalizability**: The adversary trained using RL+GNN exhibits remarkable generalization capabilities across different data distributions, including unseen graphs. As demonstrated in the "unseen" column of Tables 4-9, the trained adversary causes **greater average performance degradation** in the combinatorial player's strategies compared to the stochastic adversary (**3.38%** on average of all problems).
>
> **W2: Related Work**
>
> We appreciate your insights into the literature on zero-sum games. Based on your feedback, we have enhanced our related work (Section 2, paragraph 2), adding the following sentence and several references,
>
> “Except for DO and its variants, NE learning in zero-sum settings remains appealing in periodic games [1], polymatrix games [2], Markov games [3], etc.”
>
> **W3: Novelty and significance of existence and convergence results**
>
> Thanks for your comments. We have rewritten and highlighted novelties of the existence of NE and CCDO in Section 4 paragraph 2, and Section 5.1. Here we emphasize those novelties as follows.
>
> [The existence of NE]
>
> In the first paragraph of Section 4, Lines 220-225, we have highlighted the reason why the existence of NE of ACCES games can not be derived from matrix games and continuous games directly. In short, the elements in ACCES games violate the basic principles of the existence of NE in matrix games – finite strategies, and that in the continuous game – the continuity of the utility function on $X \times Y$.
>
> More specifically about the difficulty, in the ACCES game, the existence of NE requires the **weakly sequentially compact property** of the joint mixed strategy space and **continuity of the expected utility function**, which has not been established in the existing literature. We fill the gap by proving the two properties in ACCES games (Propositions 1 and 2), which further play a foundational role in proving the existence of NE (Theorem 1).
>
> [Novelties of convergence analysis of CCDO and CCDO-RL]
>
> * In Section 5.1 Lines 311-315, we describe the inner mechanism of the convergence of DO and its variants.  For DO and its variants (like ODO, XDO, etc.), the convergence of their algorithms mainly **relies on the finite strategy space property** (to transform the subgame into the original game by adding the best response (BR) over a finite number of iterations), which does not work for the **infinite/continuous strategy space** in ACCES games, and this fundamentally **alters the structure of convergence analysis**. Hence the first novelty is that our algorithms, CCDO and CCDOA both have the convergence guarantee in the ACCES game.
>
> * We propose the convergence analysis with approximate best responses (ABRs) and **different ABRs’ influence on the convergence**. ABRs are very commonly used in COPs due to their NP-hardness. It’s therefore critical to consider its effect on the convergence of ACCES games which wasn’t addressed before. We provide the novel convergence analysis of CCDOA\CCDO-RL and study different ABR’s influence on convergence (Theorem 3 Item 2 and Remark 2) in Section 5.4.
>
> **W4: Summarizing W1 and W3**
>
> About the motivation of the adversary and novelties of theory, please see our responses to W1 and W3.
>
> We need to bring to the attention of the reviewer that, in addition to the theoretical contributions, we also proposed a practical algorithm to solve real-world problems. This is challenging as even a sub-problem of finding the BR for the combinatorial strategy space of one player is known to be NP-hard, let alone the entire ACCES game. We bridge this gap by proposing CCDO-RL, which adopts RL as an efficient sub-routine to compute the ABRs.
>
> [1] Fiez, T. et al. (2021). Online learning in periodic zero-sum games. In Neurips, volume 34, pages 10313–10325.
>
> [2] Cai, Y. et al. (2016). Zero-sum polymatrix games: A generalization of minmax. Mathematics of Operations Research, 41(2):648–655.
>
> [3] Zhu, Y. and Zhao, D. (2020). Online minimax q network learning for two-player zero-sum markov games. IEEE Transactions on Neural Networks and Learning Systems, 33(3):1228–1241.

---

### Official Review · Reviewer_H3qj · 2024-11-04

**Soundness:** 3
**Presentation:** 4
**Contribution:** 3
**Rating:** 6
**Confidence:** 3

**Summary:**

The paper introduces a new class of asymmetric zero-sum games that features combinatorial action space for one player and an infinite compact space for the other, termed Asymmetric Combinatorial-Continuous zEro-Sum (ACCES) games. After providing its definitions and motivations, the authors prove the existence of mixed NE in ACCES games and design two algorithms to solve for NE, with proofs and analysis on their convergence. The second algorithm, which adopts RL concepts, is further validated on three instances of ACCES games and achieves positive experimental results.

**Strengths:**

+ Good presentations, particularly motivations for ACCES games.
+ The paper designs an algorithm with proven convergence guarantee called CCDOA to solve ACCES games, which extends the idea of double oracle-based solutions from zero-sum finite games.
+ The paper further develops a more practical version of CCDOA that uses RL and graph embedding techniques to find the approximate best response for each player.

**Weaknesses:**

- Experiments for evaluating CCDOA-RL are small-scale, with at most 50 nodes, which limits the practicality of the proposed algorithm. Furthermore, there were no discussions on runtimes or potential performance optimization for reducing computations.
- Wrong bibliography style (should be [author(s), year], not numbers).

**Questions:**

1. Could the authors provide some remarks on ACCES games with more than 2 or more generally N players i.e., n players with combinatorial action space, and N-n players with infinite compact space? In particular, the existence of NE and generalizability of the proposed algorithms to such settings?
2. What are the runtimes for CCDOA-RL on 50-node instances?

---

> ### Author Response · Authors · 2024-11-21
> **Weaknesses**
>
> Thank you for your feedback. Below, we address your comments point by point.
>
> **W1: Scalability and runtime discussion**
>
> These are great points. Although our conclusion section has summarized that scalability is not the focus of our current work, we agree that this is a critical aspect and hence we have provided the following further discussions in the revised version (Appendix E).
>
> **1. [Scale and runtime analysis]**
>
> In CCDO-RL, three components need to be trained or computed:
> * The **combinatorial player**’s policy. This player solves a combinatorial optimization problem (COPs) under a specific strategy of the adversary.
> * The **continuous player** (as the adversary) with an infinite continuous strategy space.
> * The computation of **Mixed Nash Equilibria** (NE) in the subgame.
>
> Next, we will analyze the computation time for each component individually theoretically and experimentally. For the experimental part, we will use the 50-node Patrolling Game (PG) scenario, which is the most challenging problem in our experiments, as an example.
>
> * The combinatorial player is trained using Graph Neural Networks (GNN) and REINFORCE to find feasible and optimal solutions for **NP-complete** COPs. This complexity requires RL to invest more time and data for effective model training. Training a stable and high-performing combinatorial model takes **26 minutes** (10000 data, 1024 batch size, 150 epochs) with the continuous player fixed.
> *  The continuous player is trained by PPO to tackle a one-step problem. It still utilizes GNN to understand graph structure. **One action per episode** leads to reduced training times compared to the combinatorial player. Training a high-performing model takes only **4 to 5 minutes** (10000 data, 1024 batch size, and 50 epochs).
> * For the NE solution, the mixed equilibria in a zero-sum game can be solved by the linear programming method which has **polynomial complexity** in the size of the game tree. From the perspective of experiments, the computational time is negligible (**less than 2s**).
>
> From the statement above, we can conclude that **more than five-sixths of the computation time** is spent training the model or strategy of the combinatorial player. Therefore, a crucial aspect of addressing the scalability issue is to enhance the speed of solving the COPs using RL.
>
> **2. [Potential ways of improvement]**
>
> We briefly discuss the following two main aspects. Note again that these methods will be in parallel to our work, and serve as future work.
>
> * COPs simplification method
>   * **The pruning method**: this one was introduced in the original scale of COPs to reduce the number of possibly useful actions. In this way, the computational burden will be decreased [1, 2].
>   * **Broken down into subproblems**: in some concrete COPs like TSP[3], and VRP [4], the originally large-scale problem can be broken down into smaller problems to solve, thereby reducing the solution difficulty.
>
> * RL algorithms
>   * **Learning Time Reduction**: increase the sampling data quality by attaining good-performance data from pre-trained RL models or heuristic algorithms on COPs (seemingly like the model-based RL).
>   * **NN Model Adjustment**: design an NN model with strong scalability which means that the trained model on small-scale problem instances can be used on large-scale ones, such as in influence maximization [5].
>
> **3. [Train CCDO-RL in small graphs, test in larger unseen graphs]**
>
> We quickly tested the CCDO-RL model (trained on 50-node graphs) on larger patrolling game scenarios. For example, on unseen 100-node and 200-node graphs (100 of each type), CCDO-RL outperformed other baselines with almost negligible test runtime.
>
> | Algorithms | 100 nodes | 200 nodes |
> |--------|--------|--------|
> | Greedy_op  | 7.7050  | 11.0117  |
> | RL with Stoc  | 7.8318  | 9.2422  |
> | CCDO-RL  | **8.4165**  | **11.0711**  |
>
> We will add more experiment results on CSP and CVRP.
>
> **W2: Wrong bibliography style**
>
> Thanks for pointing it out! We have corrected the bibliography style in the updated version.
>
> [1] Manchanda S, Mittal A, Dhawan A, et al. Learning heuristics over large graphs via deep reinforcement learning[J]. arXiv preprint arXiv:1903.03332, 2019.
>
> [2] Lauri J, Dutta S, Grassia M, et al. Learning fine-grained search space pruning and heuristics for combinatorial optimization[J]. Journal of Heuristics, 2023, 29(2): 313-347.
>
> [3] Fu Z H, Qiu K B, Zha H. Generalize a small pre-trained model to arbitrarily large tsp instances[C]// AAAI. 2021, 35(8): 7474-7482.
>
> [4] Hou Q, Yang J, Su Y, et al. Generalize learned heuristics to solve large-scale vehicle routing problems in real-time[C]// ICLR. 2023.
>
> [5] T. Chen, S. Yan, J. Guo, and W. Wu, "ToupleGDD: A Fine-Designed Solution of Influence Maximization by Deep Reinforcement Learning," in IEEE Transactions on Computational Social Systems, vol. 11, no. 2, pp. 2210-2221, April 2024.

---

> > ### Comment · Reviewer_H3qj · 2024-11-27
> >
> > I greatly appreciate the in-depth discussion from the authors regarding scalability. This has addressed my concern, and hence I will maintain my positive view of the paper.

---

> > > ### Author Response · Authors · 2024-11-29
> > >
> > > Thanks for affirming us of your positive view. We hope that these results will further improve the completeness of our paper. We will be more than happy to answer any additional questions and welcome more suggestions!

---

> ### Author Response · Authors · 2024-11-21
> **Questions**
>
> **Q1: Extension to N-player ACCES games**
>
> Excellent questions! Thank you for pointing out. We have added the N-player remark in Section 4 (analysis details added in Appendix A.2). Our concrete analysis is as follows.
>
> 1. The existence of NE (N-player ACCES games)
>
> **Our propositions and Theorem 2 can be extended to the N-player ACCES games naturally.** The key point of the existence of NE to N-player ACCES games is two fundamental properties we propose in ACCES games, weakly sequential compactness of the mixed strategy space and continuity of the expected utility function (Propositions 1 and 2), and the approximation idea by finite games. We introduce these as follows:
>
> * [Two properties] In Proposition 1, we transform the weakly sequential compactness of the joint mixed strategy space into the separability and weakly sequential compactness of each single player by Lemma 1. In Proposition 2, we scale the distance between two mixed strategies to the sum of distances between a single player’s mixed strategies while fixing other players. **According to the proof of these two propositions, they are all independent of the number of players**.
>
> * [The approximation idea by finite games] The main idea is to approximate the infinite continuous strategy space by finite grids by definitions of approximate games and essentially finite games. The idea and definitions are not limited to the two-player setting.
>
> 2. CCDO & CCDO-RL algorithms
>
> Due to the focus on the double oracle setting of our algorithms (CCDO, CCDO-RL), there is no theoretical guarantee possibly on the N-oracle setting. More potential algorithms in the multi-player RL field can be used for reference to develop the N-player ACCES game [1].
>
> **Q2: Runtimes on 50-node instances**
>
> The following time also contains intermediate variable storage (best response models in each round), and algorithm computation (training two approximate best responses and mixed NE solution). We have added the runtimes on 50-node instances in Appendix F.1. Note that since scalability is not the focus of this work, we did not pay much attention to runtime reduction. As we have discussed in the response to W1 above, there is a suite of methods that can be potentially used to reduce the runtime, the effort of which can be made in parallel with our work. Also, our initial results above show that the RL policies trained on smaller graphs (e.g., 50 nodes) can be **generalized to larger graphs (e.g., 100 and 200 nodes)**, as shown in our response to W1.
>
> * Adversarial covering salesman problem: 10h 20mins
> * Adversarial capacitated vehicle routing problem: 4h 40mins
> * Patrolling Game: 9h 6 mins
>
> [1] Zhang, Youzhi, and Bo An. "Converging to team-maxmin equilibria in zero-sum multiplayer games." International Conference on Machine Learning. PMLR, 2020.

---

> ### Author Response · Authors · 2024-11-25
> **Additional Experiment results of W1.**
>
> We have completed an additional set of experiments on the CSP problem. Similarly, we tested the CCDO-RL model (trained on 50-node graphs) on larger-scale scenarios (100-node, 200-node, and 500-node unseen graphs, averaged over 100 of each type). The results demonstrate that CCDO-RL performs better than the baselines while requiring significantly less test time compared to the heuristic algorithm.
>
> | Algorithms  | 100 nodes |  200 nodes | 500 nodes |
> | ----------- | ----------- | ------------- | -------------|
> | Heuristic   | 7.3808 (5h 46mins)  | 6.9543 (7h 16mins) | 6.7778 (9h 39mins) |
> | RL with Stoc   | 7.3376 | 9.8635 | 14.9543 |
> | CCDO-RL | **4.6134** | **4.8905** | **5.0541** |
>
> We hope our responses address your concerns, and we are happy to clarify further if needed.

---

### Author Response · Authors · 2024-11-21
**Global Response**

Dear Reviewers,

We sincerely thank all the reviewers for your constructive and valuable feedback. We are particularly encouraged by your recognition of the following strengths in our paper:
* Good presentation with clear motivations (Reviewers H3qj, ptbk, xkfi).
* Acknowledgment and Interest in the ACCES game setting (Reviewers H3qj, xkfi, and LGc6).
* Solid theoretical proofs of the existence of NE and convergence analysis (Reviewers H3qj, ptbk, xkfi, and LGc6) and their digestibility to readers (Reviewer xkfi).
* Impressive algorithm designs, i.e. both exact and approximate versions, CCDO and CCDORL (Reviewers H3qj and xkfi).
* Be valuable to the community (Reviewer xkfi) and empirical demonstrations of their effectiveness (Reviewer LGc6).

We rigorously addressed and incorporated the feedback to strengthen our work. Below are the main changes to the updated version:
* We added the scalability discussion about runtime and potential performance optimization methods in Appendix E.
* Added the discussion on the existence of NE in the $N$-player ACCES game ($n$ combinatorial players and $N-n$ continuous players) in Appendix A.2.
* Moved the original Definition 1 and Lemma 1 to the Appendix to save space for the newly added contents.
* Supplemented statements of the dynamic of ACCES games (Section 1, paragraph 4), clarification of patrolling game (Section 1, paragraph 4), literature supplement (Section 2, paragraph 2), the solving algorithm for mixed NE (Section 5.2, paragraph 2), experimental impact of Lines 6-9 in Algorithm 1 (Section 5.2, paragraph 1), and different ABRs’ influence (Section 5.3, paragraph 2).
* Additional changes in terms of notations, bibliography style, etc.

We will address each reviewer's specific comments in the respective responses.

---

### Meta-Review · Area_Chair_PXD9 · 2024-12-20

**Metareview:**

The paper studies a class of zero-sum games in which one player has a combinatorial action space and the other has a continuous, compact action space. This setting extends matrix-based games. The authors motivate the setting using the scenario of patrolling games among others. The authors prove existence of Nash Equilibrium (NE) in these games, and propose two algorithms (CCDO and CCDO-RL) to solve these games, and validate their approaches empirically. CCDO uses exact best responses, while CCDO-RL leverages reinforcement learning to compute approximate best responses, allowing for scalability in practical applications.

All reviewers recommended acceptance, with one showing strong support.

**Additional Comments On Reviewer Discussion:**

The short discussion revolved around scalability experiments and a couple of technical claims. All were resolved to the satisfaction of the reviewers, as reflected in the maintained or improved ratings.

During the discussion period, reviewers raised concerns mostly regarding scalability, novelty relative to prior work, and algorithmic details. For scalability, the authors provided runtime analyses, additional experimental results on larger problem instances, and a discussion of potential optimizations such as distributed training and pruning. Reviewer H3qj "greatly appreciate[d] the in-depth discussion from the authors regarding scalability. This has addressed [their] concern."
For algorithmic details, Reviewer LGc6 had questions regarding the termination criteria of CCDO-RL. The authors clarified that "the mixed Nash equilibrium in the ACCES game [is] solved using the support enumeration algorithm," and described the termination as based on stopping criteria in the algorithm rather than game dynamics. Reviewer LGc6 responded positively, stating, "Thank you for addressing my questions and concerns. I will raise my score to a 6."
For novelty, Reviewer LGc6 wrote: "This paper tackles a novel problem in 2p0s games by considering asymmetry in strategy space rather than the common forms of asymmetry (e.g., information asymmetry)."

---

### Decision · Program_Chairs · 2025-01-22

Accept (Poster)